# EAST: Early Action Prediction Sampling Strategy with Token Masking

**Iva Sović, Ivan Martinović, Marin Oršić**
Faculty of Electrical Engineering and Computing, University of Zagreb, Croatia
`{iva.sovic,ivan.martinovic,marin.orsic}@fer.hr`

## ABSTRACT

Early action prediction seeks to anticipate an action before it fully unfolds, but limited visual evidence makes this task especially challenging. We introduce EAST, a simple and efficient framework that enables a model to reason about incomplete observations. In our empirical study, we identify key components when training early action prediction models. Our key contribution is a randomized training strategy that samples a time step separating observed and unobserved video frames, enabling a single model to generalize seamlessly across all test-time observation ratios. We further show that joint learning on both observed and future (oracle) representations significantly boosts performance, even allowing an encoder-only model to excel. To improve scalability, we propose a token masking procedure that cuts memory usage in half and accelerates training by 2× with negligible accuracy loss. Combined with a forecasting decoder, EAST sets a new state of the art on NTU60, SSv2, and UCF101, surpassing previous best work by 10.1, 7.7, and 3.9 percentage points, respectively.

## 1 INTRODUCTION

Action recognition enables machines to identify, understand and interpret human activities in video (Bobick & Davis, 2001; Karpathy et al., 2014). Many important applications of this task require hard real-time inference in order to ensure a timely reaction or a precautionary measure. Examples include security surveillance (Wren et al., 1997), human-robot interaction (Breazeal, 2003), autonomous driving (Geiger et al., 2012), workplace safety, and other safety-critical applications. Such applications benefit from accurate predictions even before the action took place in its entirety. This state of affairs motivates a subtask known as early action prediction or early action recognition (Hu et al., 2019; Foo et al., 2022; Kong et al., 2017; Stergiou & Damen, 2023; Ryoo, 2011).

Early action recognition methods classify actions from a partially observed part of the video (Ryoo, 2011). This makes the task challenging since the model should consider upcoming future content that is inherently a multi-modal distribution (Vondrick et al., 2016; Baltrušaitis et al., 2019). Recent methods find future action cues using auxiliary methods that do not always benefit early action classification performance, such as motion forecasting (Pang et al., 2019; Liu et al., 2023), future residual forecasting (Zhao & Wildes, 2019) or modelling the possible future state using graphs (Wu et al., 2021b). Furthermore, the latest methods require separate models for each observation ratio. This requires immense training resources and complicates model deployment.

In this work, we propose EAST (Early Action prediction Sampling strategy with Token masking), an end-to-end framework that learns to predict actions from partial observations more effectively and efficiently. The core concept within EAST is a frame sampling strategy that enables training a single model for all observation ratios. During training, EAST samples partially observed (present) videos for all observation ratios, as well as full videos (future). Compared to methods that train per observation ratio models, our strategy simplifies inference and speeds up training 9× when there are 9 observation ratios. In contrast to previous methods that use auxiliary objectives, we simplify the learning objective by directly optimizing action prediction performance. Moreover, we greatly improve training efficiency by masking input patches that change the least over time. Remarkably, we find that as much as 50% of tokens can be removed without significantly degrading performance.

Token masking reduces inference time, but primarily aims efficient training: total GPU time and memory reduces by $2\times$, allowing EAST to train using two GPUs with 20GB of memory.

EAST involves three main contributions. First, we propose a framework that trains a single model based on classifications of common encoder features from dynamically sampled present and future video frames. We achieve further improvements using a forecasting decoder over present features. The proposed setup greatly improves efficiency since a single model is tested across all observation ratios. Second, we improve training efficiency by removing repetitive tokens according to visual similarity of input patches. Third, we evaluate our contributions through extensive validations on standard action classification datasets. EAST sets the new state-of-the-art across all evaluation settings for early action prediction on NTU60 (Liu et al., 2019), Something-Something V2 (Goyal et al., 2017) and UCF101 (Soomro et al., 2012). Source code is available at https://github.com/ivasovic/east.

## 2 RELATED WORK

**Action recognition** strives to interpret human activities after observing the entire video. The seminal approach by Karpathy et al. (2014) finds temporal structure by combining independent 2D convolutions, while Simonyan & Zisserman (2014) propose separate appearance and motion processing. Spatio-temporal features are naturally extracted with 3D convolutions (Ji et al., 2012; Tran et al., 2015; Lin et al., 2019; Feichtenhofer et al., 2019). These models benefit from ImageNet by repeating pre-trained 2D convolutional kernels into the temporal dimension (Deng et al., 2009; Carreira & Zisserman, 2017). However, convolutional architectures struggle with long-term spatio-temporal features and excessive model complexity (Wang et al., 2016; Feichtenhofer et al., 2019; Xie et al., 2018; Tran et al., 2015). Therefore, the most recent work favours transformer-based approaches (Piergiovanni et al., 2023; Li et al., 2023; Ryali et al., 2023; Li et al., 2022c;b;a; Tong et al., 2022; Wang et al., 2023; Srivastava & Sharma, 2024)

**ViT token removal.** Masked image modelling is an effective self-supervised pretext task (Dosovitskiy et al., 2020; He et al., 2022; Zhou et al., 2022; Gupta et al., 2023). Fortunately, masking input tokens greatly reduces training time and memory complexity. This is especially important for long videos due to quadratic complexity of self-attention. VideoMAE and MAE-ST extend masked image modelling to video using a very high masking ratio of spatio-temporal cubes known as tubelets (Tong et al., 2022; Feichtenhofer et al., 2022; Piergiovanni et al., 2023; He et al., 2022). VideoMAE V2 extends the masking to the image reconstruction decoder (Wang et al., 2023).

Token masking also benefits supervised training. NaViT trains on combinations of entire and subsampled image tokens (Dehghani et al., 2023). DynamicViT hierarchically prunes redundant tokens in an online manner (Rao et al., 2021). EVEREST removes tokens from uninformative frames and redundant patches (Hwang et al., 2024). Other approaches reduce tokens by similarity (Liang et al., 2022; Bolya et al., 2023; Fayyaz et al., 2022; Yin et al., 2022; Haurum et al., 2023; Choudhury et al., 2024). DTEM decouples feature representation learning from token merging (Lee & Hong, 2024). Our token masking optimizes training while retaining early action prediction accuracy.

**Action anticipation** methods predict future actions before they begin. A prominent approach anticipates future frames in unlabeled video (Vondrick et al., 2016). AVT anticipates actions with a per-frame encoder and a supervised causal decoder (Girdhar & Grauman, 2021). Fernando & Herath (2021) supervise feature forecasting using Jaccard similarity between observed and future features. Our approach does not optimize a feature similarity measure. Instead, we use a video-specific encoder and include a novel discriminative loss to the training objective.

**Early action prediction** considers methods that classify partially observed videos (Wang et al., 2019). Early work considers action dynamics from a bag of visual words (Ryoo, 2011), or using LSTM memory that records early observations (Hochreiter & Schmidhuber, 1997; Kong et al., 2018a). ERA finds subtle early differences between actions with a mixture of experts (Foo et al., 2022; Jacobs et al., 1991). IGGNN+LSTGN models spatio-temporal object relationships using features around bounding box detections (Wu et al., 2021a). MSRNN uses soft regression on early frame features to account for future action uncertainty (Hu et al., 2019). Early-ViT learns action prototypes to regularize partial video representations (Camporese et al., 2023). TemPr processes a temporal feature cascade using consensus between transformer towers (Stergiou & Damen, 2023).

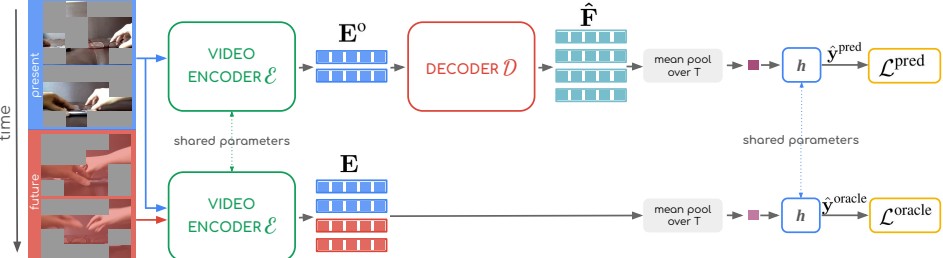

Figure 1: EAST uses both present and future frames in training. ViT encoder $\mathcal{E}$ processes the observed frames (blue) and the entire video (blue and red). Decoder $\mathcal{D}$ observes present features $\mathbf{E}^o$ and forecasts future features $\hat{\mathbf{F}}$. $h$ classifies actions from decoder features and oracle encoder features $\mathbf{E}$. We optimize both classification scores but only use $\hat{\mathbf{y}}_{\text{pred}}$ during inference.

Similar to us, some early action prediction methods guide anticipative future representations by training with entire videos. DBDNet trains a Bi-LSTM that bidirectionally reconstructs present and future motion (Pang et al., 2019). LST-GCN models spatio-temporal evolution of object relationships using graph convolutional networks (Wu et al., 2021b). AA-GAN forecasts future representations by leveraging optical flow (Gammulle et al., 2019). Furthermore, AA-GAN enhances future representations using adversarial training, where a discriminator discerns between generated and oracle future features. Similarly, an action recognition teacher can supervise the student that receives only early video frames (Wang et al., 2019). DeepSCN starts by learning enriched features that minimize the discrepancy between partial observations and full observations (Kong et al., 2017). Consequently, it learns an SVM model to classify enriched partial features into categorical actions. Zhao & Wildes (2019) propose to forecast the future residuals with a Kalman filter and then recursively integrate them into feature representations of unobserved frames that are separately classified. Unlike all previous approaches, we express the recognition of partially forecasted and completely observed sequences purely using discriminative losses. The classifier infers from pre-trained encoder features and also from forecasted decoder features within end-to-end training. Most importantly, our training strategy samples observation ratios when preparing training samples. This procedure enables good generalization with an arbitrary observation ratio.

## 3 METHOD

Early action prediction involves making predictions while observing a fraction of the video. There are $T_d$ video frames. The observation ratio $\rho \in \langle 0, 1 \rangle$ controls the fraction of observed (present) frames. Therefore, the model predicts early actions based on frames in $[0, \rho \cdot T_d\rangle$. In training, the model has access to all $T_d$ frames and one-hot annotations $\mathbf{y}$. In inference, the model makes predictions based on the first $\rho \cdot T_d$ frames, where standard practice evaluates using $\rho$ from $0.1$ to $0.9$ in increments of $0.1$. We follow this setting and apply a unified model to all observation ratios.

There are three main parts in EAST. First, our frame sampling strategy enables training a single model at all observation ratios by sampling observed and unobserved clips. Second, we optimize an objective that enforces correct predictions from observed frames and also from entire clips. Third, we reduce the video transformer training memory using token masking based on visual repetitiveness, without compromising accuracy. Next, we explain the details of these steps, beginning with the most important: frame sampling.

### 3.1 SAMPLING STRATEGY FOR TRAINING EARLY ACTION PREDICTION

In training, we randomly sample $\rho \in \{0.1, 0.2, 0.3, ..., 0.9\}$. Using $\rho$, we collect $T$ observed frames $\mathbf{V}^o \in \mathbb{R}^{T \times H \times W \times C}$ and $T$ unobserved frames $\mathbf{V}^u \in \mathbb{R}^{T \times H \times W \times C}$ so that $\mathbf{V}^o$ temporally precedes $\rho \cdot T_d$ and $\mathbf{V}^u$ succeeds $\rho \cdot T_d$. The sampled clip $\mathbf{V} = \mathbf{V}^o \| \mathbf{V}^u$ consists of $2T$ evenly spaced frames. We ensure that the final frame in $\mathbf{V}^o$ and the first frame in $\mathbf{V}^u$ are adjacent frames in the original video, avoiding temporal distortion in training samples. Randomizing $\rho$ in training enables the model to adapt to variable temporal context length.

Although conceptually simple, this training setup is essential for early action prediction. Our preliminary experiments suggest that training at fixed observation ratios produces models that are suboptimal at other values of $\rho$. Such setup would require training specialized models and hinder real-world applications. Furthermore, we find that off-the-shelf action recognition models fail on early action prediction since they depend on the full context.

## 3.2 FORECASTING WITH MAE FEATURES

**Encoder.** The encoder architecture closely follows Vision Transformers (ViT) with spatio-temporal positional encodings to account for video (Vaswani et al., 2017; Dosovitskiy et al., 2020). The ViT-based encoder $\mathcal{E}$ consists of tokenizer $\mathcal{T}$ and transformer encoder $\mathcal{V}$. Concretely:

$$\mathcal{E}: \mathbb{R}^{T \times H \times W \times C} \to \mathbb{R}^{N_t \times F}, \quad \mathcal{E}(\mathbf{V}^{\mathrm{o}}) = \mathcal{V} \circ \mathcal{T}(\mathbf{V}^{\mathrm{o}}) = \mathbf{E}^{\mathrm{o}}. \tag{1}$$

The model processes input frames with $C = 3$ RGB channels. The encoder extracts tokens with $F$ features. Tokenizer $\mathcal{T}$ splits the input clip frames into $N_t = \frac{THW}{p^2 d}$ non-overlapping tubelets of size $d \times p \times p$, where $p = 16$ and $d = 2$ determine the spatial and temporal tubelet size, respectively. Spatio-temporal information is added to tokens via sin-cos embeddings. The transformer encoder $\mathcal{V}: \mathbb{R}^{N_t \times F} \to \mathbb{R}^{N_t \times F}$ extracts features from the input video clip.

**Decoder** $\mathcal{D}$ forecasts future features $\hat{\mathbf{F}}$ given $\mathbf{E}^{\mathrm{o}}$:

$$\mathcal{D}: \mathbb{R}^{N_t \times F} \to \mathbb{R}^{\frac{T}{d} \times F}, \quad \mathcal{D}(\mathbf{E}^{\mathrm{o}}) = \mathcal{F} \circ P_s(\mathbf{E}^{\mathrm{o}}) = \hat{\mathbf{F}}. \tag{2}$$

The encoded present features $\mathbf{E}^{\mathrm{o}} = \mathcal{E}(\mathbf{V}^{\mathrm{o}})$ are input to spatial average pooling $P_s : \mathbb{R}^{N_t \times F} \to \mathbb{R}^{\frac{T}{d} \times F}$ that produces a mean token for each time step. Decoder module $\mathcal{F}$ processes $\frac{T}{d}$ present tokens and forecasts $\frac{T}{d}$ future tokens.

We produce a strong baseline by setting $\mathcal{F}$ to an identity mapping, effectively making the method decoder-free. We further evaluate the decoder design with two distinct architectures: i) autoregressive and ii) direct transformer. Autoregressive formulation of $\mathcal{F}$ observes $\mathbf{E}^{\mathrm{o}}$ and forecasts tokens with causal inference. Direct inference concatenates $\mathbf{E}^{\mathrm{o}}$ with additional [MASK] tokens, and performs a single forward pass through a full attention transformer. Based on the validation results, we set the direct 4-layer transformer as $\mathcal{F}$ within EAST.

## 3.3 COMPOUND FORECASTING LOSS

Figure 1 contains a diagram of the training setup. The partially observed clip $\mathbf{V}^{\mathrm{o}}$ is classified using:

$$\hat{\mathbf{y}}^{\mathrm{pred}} = h \circ P_t \circ \mathcal{D} \circ \mathcal{E}(\mathbf{V}^{\mathrm{o}}). \tag{3}$$

Here, $h$ produces early action classification logits using a linear layer, and $P_t$ denotes mean pooling.

The encoder features should be both discriminative and contain cues about future features. To achieve this in training, we perform an additional forward pass through $\mathcal{E}$. We compute oracle encoder features using the entire sampled clip: $\mathbf{E} = P_s \circ \mathcal{E}(\mathbf{V})$. Consequently, the common classifier produces two sets of classification logits. The first set $\hat{\mathbf{y}}^{\mathrm{pred}} = h \circ P_t(\hat{\mathbf{F}})$ contains early action classification logits obtained by forecasting from $\mathbf{E}^{\mathrm{o}}$, whereas the second vector $\hat{\mathbf{y}}^{\mathrm{oracle}} = h \circ P_t(\mathbf{E})$ contains classification logits for the entire sampled video clip.

We train the model from end-to-end to minimize the average compound loss $\mathcal{L}$ that sums negative log-likelihoods:

$$\mathcal{L} = \mathcal{L}^{\mathrm{pred}} + \mathcal{L}^{\mathrm{oracle}} = \mathcal{L}_{\mathrm{NLL}}(\hat{\mathbf{y}}^{\mathrm{pred}}, \mathbf{y}) + \mathcal{L}_{\mathrm{NLL}}(\hat{\mathbf{y}}^{\mathrm{oracle}}, \mathbf{y}). \tag{4}$$

This formulation is intuitive when considering the loss gradients. Gradients through $\mathcal{L}^{\mathrm{pred}}$ directly optimize early action prediction and enforce discriminative features in both the encoder and the decoder. Gradients through $\mathcal{L}^{\mathrm{oracle}}$ yield discriminative features when observing a full video. We find that the combination of the two losses yields the best early action prediction.

## 3.4 EFFICIENT TOKEN MASKING

To reduce computational costs of attention layers, we propose to mask temporally repeating tokens. The proposed token masking strategy has been inspired by the Moravec corner detector (Harris

& Stephens, 1988). This masking strategy primarily reduces the training memory footprint, making EAST suitable for training on more affordable GPU setups. Also, the proposed masking reduces inference time by reducing the number of input tokens.

We find repeating tokens according to L1 patch distances throughout time (Choudhury et al., 2024). Tubelet volume is set by patch size $p$ and tubelet size $d$. Thus, we extract vectorized non-overlapping patches $\mathbf{p}_{t,i,j}$ using:

$$\mathbf{p}_{t,i,j} = \mathbf{V}_{[td:td+d,ip:ip+p,jp:jp+p]} \tag{5}$$

In each frame $t$ from video $\mathbf{V}$, we rank each tubelet according to pixel distance from the last patch in the next tubelet:

$$r_{t,i,j}(\mathbf{V}) = \|\mathbf{p}_{t,i,j[0]} - \mathbf{p}_{t+1,i,j[d-1]}\|_1 \tag{6}$$

Note that we compare the first and the last patch since size $d > 1$. Finally, we keep the highest ranking tubelets using:

$$\mathcal{M}_k^{\mathrm{d}}(\mathbf{V}) = \{\mathbf{p}_{t,i,j} : r_{t,i,j}(\mathbf{V}) \geq r_{i,j}^k\} \tag{7}$$

$r_{i,j}^k$ denotes ranking for the $k$-th quantile at spatial position $(i, j)$. We set $k = 50\%$ in our experiments and apply token masking when computing both present and oracle features. Note that masking with $k$ removes the same number of tokens from each spatial position. In other words, we halve the number of input tubelets at each spatial position. We refer to $\mathcal{M}_k^{\mathrm{d}}$ as difference masking. In training, we apply $\mathcal{M}_k^{\mathrm{d}}$ independently to $\mathbf{V}^{\mathrm{o}}$ and $\mathbf{V}^{\mathrm{u}}$ to prevent information leakage. Note that we use feature extractors pre-trained with MAE (He et al., 2022; Tong et al., 2022). Therefore, the encoder is unaffected by masking since there is no distribution shift compared to MAE pre-training.

## 4 EXPERIMENTS

We compare EAST with related methods on four datasets used in the early action prediction setup: Something-Something, versions v2 and sub21 (Goyal et al., 2017), NTU RGB+D (Liu et al., 2019), UCF101 (Soomro et al., 2012) and EK-100 (Damen et al., 2022). We also include experiments that measure the influence of proposed components. Refer to the supplement for detailed insights.

### 4.1 DATASETS

**Something-Something v2 (SSv2)** is a large-scale video dataset primarily used for action recognition. The dataset consists of $220\,\mathrm{k}$ video samples and $174$ classes. There are $169\,\mathrm{k}$ training videos and $20\,\mathrm{k}$ validation videos, whereas the remaining unlabeled videos are used for testing. SSsub21 is a Something-Something subset typically used in early action prediction evaluation. It contains $21$ action classes across $11\,\mathrm{k}$ videos. We include experiments on SSsub21 to compare with most previous methods. We also include results on the full SSv2 dataset.

**NTU RGB+D** dataset consists of $60$ action classes and has $57\,\mathrm{k}$ $1920{\times}1080$ RGB videos. Most samples also include depth maps, infrared frames and skeletal keypoints. We use only the RGB modality to train EAST. Following previous work, we use cross-subject evaluation in our experiments (Ma et al., 2016; Kong et al., 2017; Hu et al., 2019; Wang et al., 2019; Pang et al., 2019; Stergiou & Damen, 2023). There are $20$ subjects in both training and evaluation sets. This split provides $40.3\,\mathrm{k}$ training examples and $16.5\,\mathrm{k}$ testing examples.

**UCF101** is a small scale dataset that consists of approximately $13\,\mathrm{k}$ videos with $101$ action classes. Videos are divided into $9.5\,\mathrm{k}$ training and $3.5\,\mathrm{k}$ validation videos. The video resolution is $320{\times}240$ with a frame rate of $25$ FPS.

**Epic-Kitchens-100** (EK-100) is a large collection of egocentric videos consisting of 90 thousand action segments. The taxonomy consists of *verb* and *noun* category groups that determine the *action* group. The evaluation considers classification in all three category groups.

### 4.2 IMPLEMENTATION AND TRAINING DETAILS

Unless otherwise stated, we use the ViT-B/16 video encoder pre-trained on K400 using Video-MAE (Kay et al., 2017; Tong et al., 2022). Decoder $\mathcal{F}$ is also initialized from VideoMAE pre-training. This yields slight improvements over random init. We train the entire model end-to-end.

We sample $T = 8$ frames for both $\mathbf{V}^{\mathrm{o}}$ and $\mathbf{V}^{\mathrm{u}}$, and train on random $224 \times 224$ crops using MixUp augmentations (Zhang et al., 2018). We use AdamW with base learning rate $1 \times 10^{-3}$ and weight decay 0.05. The base learning rate is scaled by $\frac{\text{batch size}}{256}$ and decayed using the cosine rule (Loshchilov & Hutter, 2016). We set the batch size to 96 in SSv2, NTU60 and EK-100 experiments. In SSsub21 and UCF101 experiments, we set the batch size to 128. We train on SSv2 and EK-100 for 40 epochs, and on NTU60 and UCF101 for 50 and 100 epochs, respectively. We report results from a single training run that uses a fixed random seed. We express the computational complexity of a forward pass over a single training example. We measure this complexity as the number of floating point operations (TFLOP) using DeepSpeed (Rasley et al., 2020). We train with MixUp, therefore, our measurements reflect the complexity of processing both augmentations.

We use Nvidia RTX A6000 GPUs. All our experiments use FlashAttention optimizations that saves memory by recomputing the attention matrix in backpropagation (Dao et al., 2022). In addition to the memory savings from FlashAttention, applying $\mathcal{M}^{\mathrm{d}}_{k=0.5}$ further reduces training memory by $2\times$. The proposed training requires only $2 \times$ A6000 GPUs. Unless otherwise stated, we set $k = 0.5$ and perform the same masking in training and inference.

Since most previous work did not publish source code, evaluation details are not fully disclosed. We propose a unified protocol for early action prediction via minimal adaptations of action recognition evaluation. We apply a single model across all nine observation ratios and report top-1 accuracy. The model is agnostic to the testing observation ratio and processes present frames only. We do not subsample features pre-computed from entire videos since that would lead to unfair leak of information. We perform spatial multi-crop inference (Feichtenhofer et al., 2019). On NTU60 and SSv2, we average predictions when sliding across temporal dimension (Wang et al., 2016).

## 4.3 COMPARISON WITH THE STATE OF THE ART

**NTU60.** Table 1 shows results on the NTU60 dataset. The first section includes methods that process multiple modalities (skeletal keypoints or depth). The second section presents methods that only use RGB input frames. EAST surpasses all methods while using only RGB inputs. The average improvement over TemPr is $6.8$ pp, with the highest improvement of $19.2$ pp at $\rho = 0.3$.

Table 1: Comparison with previous work on the NTU60 dataset. We show top-1 accuracy (%) over different observation ratios. * denotes reproductions by Wang et al. (2019). We highlight input modalities as video (RGB), depth (D) and human keypoints (KP). The best results are in **bold**.

| Method | Modality | | | Observation ratio $\rho$ | | | | |
| --- | --- | --- | --- | --- | --- | --- | --- | --- |
| | RGB | D | KP | 0.1 | 0.3 | 0.5 | 0.7 | 0.9 |
| MSRNN (Sadegh Aliakbarian et al., 2017) | ✓ | ✓ | | 15.2 | 29.5 | 51.6 | 63.9 | 68.9 |
| TS (Wang et al., 2019) | ✓ | ✓ | | 27.8 | 46.3 | 67.4 | 77.6 | 81.5 |
| DBDNet (Pang et al., 2019) | ✓ | ✓ | ✓ | 28.0 | 47.3 | 68.5 | 78.5 | 81.6 |
| RankLSTM* (Ma et al., 2016) | ✓ | | | 11.5 | 25.7 | 48.0 | 61.0 | 66.1 |
| DeepSCN* (Kong et al., 2017) | ✓ | | | 16.8 | 30.6 | 48.8 | 58.2 | 60.0 |
| TemPr (Stergiou & Damen, 2023) | ✓ | | | 29.3 | 50.2 | 70.1 | 78.8 | 84.2 |
| EAST | ✓ | | | **31.2** | **69.4** | **86.2** | **87.9** | **87.9** |

Table 2: Comparison with the state-of-the-art results on SSv2 dataset. We show top-1 accuracy (%) over different observation ratios. The best results are in **bold**.

| Method | Observation ratio $\rho$ | | | | | | | | | TFLOP |
| --- | --- | --- | --- | --- | --- | --- | --- | --- | --- | --- |
| | 0.1 | 0.2 | 0.3 | 0.4 | 0.5 | 0.6 | 0.7 | 0.8 | 0.9 | |
| RACK (Liu et al., 2023) | - | 11.9 | - | 15.0 | - | - | - | 23.0 | - | - |
| Early-ViT (Camporese et al., 2023) | 22.7 | 27.8 | 33.6 | 40.5 | 48.0 | 53.9 | 58.5 | 61.5 | 63.0 | - |
| TemPr (Stergiou & Damen, 2023) | 20.5 | - | 28.6 | - | 41.2 | - | 47.1 | - | - | 0.5 |
| EAST | **25.6** | **30.1** | **34.5** | **41.6** | **49.0** | **55.2** | **59.4** | **63.0** | **64.0** | 0.5 |

**SSv2.** Table 2 presents our results on the Something-Something v2 dataset, where EAST sets the new state of the art. Average result improvement over TemPr is $28.3$ pp. For observation ratios

Table 3: Comparison with the state-of-the-art results on SSsub21. We present top-1 (%) accuracy. * refers to results that are presented by Stergiou & Damen (2023). The best results are in **bold**.

| Method | Observation ratio $\rho$ | | | | | |
|---|---|---|---|---|---|---|
| | 0.1 | 0.2 | 0.3 | 0.5 | 0.7 | 0.9 |
| MS-LSTM*(Sadegh Aliakbarian et al., 2017) | 16.9 | 16.6 | 16.8 | 16.7 | 16.9 | 17.1 |
| MSRNN* (Sadegh Aliakbarian et al., 2017) | 20.1 | 20.5 | 21.1 | 22.5 | 24.0 | 27.1 |
| mem-LSTM* (Kong et al., 2018a) | 14.9 | 17.2 | 18.1 | 20.4 | 23.2 | 24.5 |
| GGN (Wu et al., 2021b) | 21.2 | 21.5 | 23.3 | 27.7 | 30.2 | 30.6 |
| IGGN (Wu et al., 2021a) | 22.6 | - | 25.0 | 28.3 | 32.2 | 34.1 |
| TemPr (Stergiou & Damen, 2023) | 28.4 | 34.8 | 37.9 | 41.3 | 45.8 | 48.6 |
| Early-ViT (Camporese et al., 2023) | 32.1 | 35.5 | 40.3 | 52.4 | 60.7 | 62.2 |
| EAST | **40.8** | **44.7** | **51.2** | **66.4** | **75.8** | **79.3** |

Table 4: Comparison with the state-of-the-art results on the UCF101 dataset. We show top-1 accuracies (%) over different observation ratios. * denotes results reproduced by Kong et al. (2017). TemPr entry with † presents original paper results that are irreproducible using the published source code. ‡ represents our corrected reproduction.

| Method | Observation ratio $\rho$ | | | | | | | | |
|---|---|---|---|---|---|---|---|---|---|
| | 0.1 | 0.2 | 0.3 | 0.4 | 0.5 | 0.6 | 0.7 | 0.8 | 0.9 |
| MSSC* (Cao et al., 2013) | 34.1 | 53.8 | 58.3 | 57.6 | 62.6 | 61.9 | 63.5 | 64.3 | 62.7 |
| MTSSVM* (Kong et al., 2014) | 40.1 | 72.8 | 80.0 | 82.2 | 82.4 | 83.2 | 83.4 | 83.6 | 83.7 |
| DeepSCN (Kong et al., 2017) | 45.0 | 77.7 | 83.0 | 85.4 | 85.8 | 86.7 | 87.1 | 87.4 | 87.5 |
| MSRNN (Sadegh Aliakbarian et al., 2017) | 68.0 | 87.2 | 88.2 | 88.8 | 89.2 | 89.7 | 89.9 | 90.3 | 90.4 |
| mem-LSTM (Kong et al., 2018a) | 51.0 | 81.0 | 85.7 | 85.8 | 88.4 | 88.6 | 89.1 | 89.4 | 89.7 |
| AAPNET (Kong et al., 2018b) | 59.9 | 80.4 | 86.8 | 86.5 | 86.9 | 88.3 | 88.3 | 89.9 | 90.9 |
| RGN-KF (Zhao & Wildes, 2019) | 83.3 | 85.2 | 87.8 | 90.6 | 91.5 | 92.3 | 92.0 | 93.0 | 92.9 |
| DBDNet (Pang et al., 2019) | 82.7 | 86.6 | 88.4 | 89.7 | 90.6 | 91.1 | 91.7 | 91.9 | 92.0 |
| AA-GAN (Gammulle et al., 2019) | - | 84.2 | - | - | 85.6 | - | - | - | - |
| TS (Wang et al., 2019) | 83.3 | 87.1 | 88.9 | 89.9 | 90.9 | 91.0 | 91.3 | 91.2 | 91.3 |
| GGNN (Wu et al., 2021b) | 84.1 | 88.5 | 89.8 | - | 90.9 | - | 91.4 | - | 91.8 |
| IGGN (Wu et al., 2021a) | 80.2 | - | 89.8 | - | 92.9 | - | 94.1 | - | 94.4 |
| JVS (Fernando & Herath, 2021) | - | 91.7 | - | - | - | - | - | - | - |
| ERA (Foo et al., 2022) | 89.1 | - | 92.4 | - | 94.2 | - | 95.5 | - | 95.7 |
| RACK (Liu et al., 2023) | 87.6 | 87.6 | 89.4 | - | - | - | - | - | - |
| Early-ViT (Camporese et al., 2023)$_{\text{MoViNet}}$ | 87.2 | 90.1 | 91.7 | 92.2 | 92.9 | 93.4 | 93.6 | 93.5 | 93.5 |
| TempPr† (Stergiou & Damen, 2023)$_{\text{MoViNet}}$ | 88.6 | 93.5 | 94.9 | 94.9 | 95.4 | 95.2 | 95.3 | 96.6 | 96.2 |
| TempPr‡ (Stergiou & Damen, 2023)$_{\text{MoViNet}}$ | 85.1 | - | 90.4 | - | 92.5 | - | 92.8 | - | 93.2 |
| EAST$_{\text{VideoMAE}}$ | 88.6 | 91.4 | 92.2 | 93.1 | 93.4 | 93.6 | 93.7 | 93.8 | 93.8 |
| EAST$_{\text{MoViNet}}$ | **91.3** | **93.2** | **93.8** | **94.7** | **95.5** | **96.1** | **96.4** | **96.5** | **96.5** |

$\rho = 0.1$, $\rho = 0.3$, $\rho = 0.5$ and $\rho = 0.7$ we improve the results by 5.1 pp, 5.9 pp, 7.8 pp and 12.3 pp, respectively. Unlike TemPr, we train all parameters end-to-end while achieving similar TFLOP complexity. Furthermore, the results highlight the benefits of our proposed sampling strategy. Note that we evaluate a single model whereas TemPr trains a special model for each observation ratio. Therefore, TemPr requires the observation ratio during model inference, while our model is entirely agnostic to the observation ratio. Finally, Table 3 presents results on SSsub21, where the average improvement over TemPr is 22.7 pp across all observation ratios.

**UCF101.** We present our UCF101 results in Table 4. Since TemPr uses a MoViNet (Kondratyuk et al., 2021) encoder pretrained on K600 (Carreira et al., 2018), we include results with the same backbone. These results highlight the benefits of training under our proposed framework. EAST with MoViNet sets the new state-of-the-art results for all observation ratios on UCF101. Average result improvements over ERA and TemPr are 1.3 pp and 3.9 pp, respectively. These results show that our method does not depend on the backbone, and that improvements come from the proposed training strategy.

**EK-100.** We train and evaluate EAST on EK-100 using two classification heads, one for verb and one for noun. We use evaluation scripts provided by the official EK-100 repository. The results are presented in Table 5. EAST outperforms TemPr at low observation ratios but is limited by the ViT encoder accuracy at high observation ratios. TemPr uses the SlowFast backbone that achieves top-1 action recognition accuracy of 38.5 %, 50.0 % and 65.6 % on *all action*, *all noun* and *all verb*, respectively. This outperforms VideoMAE ViT-B that achieves 33.7 %, 42.7 % and 65.1 % top-1 accuracy on *all action*, *all noun* and *all verb*, respectively. The results strengthen our contributions since EAST significantly outperforms TemPr on low-observation ratios.

Table 5: Top-1 early action prediction accuracy on EK-100 under different observation ratios $\rho$.

| | All Action | | | | | All Noun | | | | | All Verb | | | | |
|---|---|---|---|---|---|---|---|---|---|---|---|---|---|---|---|
| $\rho$ | 0.1 | 0.2 | 0.3 | 0.5 | 0.7 | 0.1 | 0.2 | 0.3 | 0.5 | 0.7 | 0.1 | 0.2 | 0.3 | 0.5 | 0.7 |
| Tempr | 7.4 | 9.8 | 15.4 | **28.9** | **37.3** | 22.8 | 25.5 | 32.3 | **43.4** | **49.2** | 21.4 | 22.5 | 34.6 | 54.2 | **63.8** |
| EAST | **20.4** | **23.3** | **25.4** | 28.1 | 29.7 | **31.1** | **34.0** | **35.5** | 38.2 | 39.9 | **47.2** | **52.1** | **55.0** | **58.4** | 59.5 |

Results on all four datasets showcase that EAST generalizes in both small and large-scale datasets. Most importantly, we train one model for all observation ratios. We found that training separate models for each $\rho$ does not yield any accuracy improvements but requires $9\times$ more training time.

## 4.4 ABLATIONS AND ANALYSES

We validate EAST on SSv2, unless otherwise specified.

**Overlooked baseline EAST$_\mathcal{E}$.** Table 6 presents early action classification performance when the VideoMAE ViT-B/16 model is trained on entire action classification sequences. EAST$_\mathcal{E}$ trains the same model but uses our proposed sampling. Unlike EAST, EAST$_\mathcal{E}$ trains using $\mathcal{L}^{\text{pred}}$ only and does not have a decoder ($\mathcal{F}$ is identity). When comparing EAST$_\mathcal{E}$ to VideoMAE, there is a noticeable difference for smaller observation ratios. For $\rho = 0.1$, Video-MAE achieves only 9.9 % accuracy compared to an encoder-only EAST$_\mathcal{E}$ which achieves 23.9 %. Accuracy differences are less prominent at higher observation ratios. This is expected since high observation ratios include more context. Never-

Table 6: Top-1 SSv2 accuracy over all observation ratios. VideoMAE denotes pre-trained ViT-B/16 model performance finetuned for action classification. EAST$_\mathcal{E}$ trains ViT-B/16 with our proposed sampling without a decoder.

| | Observation ratio $\rho$ | | | | | | | | |
|---|---|---|---|---|---|---|---|---|---|
| Method | 0.1 | 0.2 | 0.3 | 0.4 | 0.5 | 0.6 | 0.7 | 0.8 | 0.9 |
| VideoMAE | 9.9 | 14.8 | 20.3 | 29.4 | 39.5 | 49.2 | 52.2 | 61.5 | 63.4 |
| EAST$_\mathcal{E}$ | 23.9 | 28.3 | 32.7 | 39.1 | 46.1 | 56.0 | 56.5 | 59.6 | 60.7 |

theless, the results indicate the critical impact of the appropriate sampling strategy. We establish a new early action prediction baseline that clearly outperforms the current state of the art on SSv2, achieving an average improvement of 5.4 pp.

**Token masking and computational efficiency.** Table 7 validates our token masking method $\mathcal{M}_k^{\text{d}}$ on NTU60. We compare $\mathcal{M}_k^{\text{d}}$ to random masking $\mathcal{M}_k^{\text{rand}}$ from VideoMAE and Running Cell masking $\mathcal{M}_k^{\text{MAR}}$ from MAR (Qing et al., 2023). We measure performance using three different masking ratios $k \in \{25\%, 50\%, 75\%\}$. The proposed difference masking outperforms other masking across tested masking ratios. This highlights the importance of retaining patches that contain most appearance change. We measure peak training memory (GB) for batch size 24 and count TFLOPs for a forward pass given one training example. As expected, masking $k = 0.75$ of patches is most efficient, but it does not achieve state-of-the-art results. Although $\mathcal{M}_{k=0.25}^{\text{d}}$ model achieves highest accuracy, we choose $\mathcal{M}_{k=0.5}^{\text{d}}$ since this setup offers best balance between efficiency and performance.

**Encoder-only vs encoder-decoder.** The first row in Table 8 shows the accuracy of an encoder-only baseline. This corresponds to the EAST$_\mathcal{E}$ entry from Table 6. The second row in Table 8 shows that training encoder-only EAST$_\mathcal{E}$ using both $\mathcal{L}^{\text{pred}}$ and $\mathcal{L}^{\text{oracle}}$ gains additional 1.5 pp. An encoder-decoder model trained using $\mathcal{L}^{\text{pred}}$ and $\mathcal{L}^{\text{oracle}}$ yields EAST, improving the average accuracy by 0.6 pp over EAST$_\mathcal{E}$. Training without $\mathcal{L}^{\text{oracle}}$ decreases results in both encoder-only and encoder-decoder setups. The results highlight the benefits of training using the proposed compound loss in both cases.

Table 7: Average NTU60 top-1 accuracy using difference masking $\mathcal{M}^{\mathrm{d}}$, Running Cell masking $\mathcal{M}^{\mathrm{MAR}}$ and random masking $\mathcal{M}^{\mathrm{rand}}$. $k$ denotes the percentage of masked tokens. We report training complexity with one example and peak training memory with batch size 24.

| Masking | avg. acc. | | TFLOP | peak mem. (GB) |
|---|---|---|---|---|
| $\mathcal{M}^{\mathrm{rand}}_{k=0.75}$ | 64.0 | ⌐ | | |
| $\mathcal{M}^{\mathrm{MAR}}_{k=0.75}$ | 66.3 | **+2.3** | 0.2 | 10.4 |
| $\mathcal{M}^{\mathrm{d}}_{k=0.75}$ | 71.9 | **+7.9** | | |
| $\mathcal{M}^{\mathrm{rand}}_{k=0.5}$ | 71.3 | ⌐ | | |
| $\mathcal{M}^{\mathrm{MAR}}_{k=0.5}$ | 72.4 | **+1.1** | 0.5 | 19.2 |
| $\mathcal{M}^{\mathrm{d}}_{k=0.5}$ | 74.3 | **+3.0** | | |
| $\mathcal{M}^{\mathrm{rand}}_{k=0.25}$ | 73.4 | ⌐ | | |
| $\mathcal{M}^{\mathrm{MAR}}_{k=0.25}$ | 73.3 | **-0.1** | 0.8 | 27.9 |
| $\mathcal{M}^{\mathrm{d}}_{k=0.25}$ | 75.2 | **+1.8** | | |
| no mask | 75.1 | | 1.1 | 36.7 |

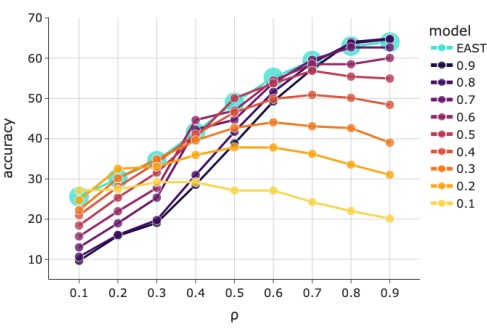

Figure 2: Comparison between EAST and models trained for a single observation ratio on SSv2. Each line denotes accuracy of a different model at each observation ratio. Training a single model with EAST comes near the accuracy of models trained for a particular $\rho$, Note that specialized models fail at observation ratios they were not trained for.

Table 8: Contributions of the proposed losses and modules to SSv2 validation accuracy. $\mathcal{D}$ denotes the choice of the decoder. `id` denotes a model where decoder $\mathcal{D}$ is set to the identity mapping. Our encoder-only approach already surpasses the previous state-of-the-art. Joint $\mathcal{L}^{\mathrm{pred}}$ and $\mathcal{L}^{\mathrm{oracle}}$ optimization benefits both encoder-only and encoder-decoder models.

| $\mathcal{L}^{\mathrm{oracle}}$ | $\mathcal{L}^{\mathrm{pred}}$ | $\mathcal{D}$ | $\rho{=}0.1$ | $\rho{=}0.2$ | $\rho{=}0.3$ | $\rho{=}0.4$ | $\rho{=}0.5$ | $\rho{=}0.6$ | $\rho{=}0.7$ | $\rho{=}0.8$ | $\rho{=}0.9$ | avg |
|---|---|---|---|---|---|---|---|---|---|---|---|---|
| | ✓ | id | 23.9 | 28.3 | 32.7 | 39.1 | 46.1 | 56.0 | 56.5 | 59.6 | 60.7 | 44.8 |
| ✓ | ✓ | id | 25.3 | 29.4 | 33.6 | 40.3 | 48.0 | 54.5 | 59.2 | 62.6 | 63.9 | 46.3 |
| | ✓ | ✓ | 26.1 | 30.4 | 34.5 | 41.2 | 48.3 | 54.1 | 58.2 | 61.0 | 61.8 | 46.2 |
| ✓ | ✓ | ✓ | 25.6 | 30.1 | 34.5 | 41.6 | 49.0 | 55.2 | 59.4 | 63.0 | 64.0 | 46.9 |

**Choice of the decoder.** AVT (Girdhar & Grauman, 2021) suggests that autoregressive prediction is natural in modelling temporal action progression for action anticipation. However, our findings in Table 9 a) show that forecasting by direct decoding outperforms autoregressive decoding by an average of $0.6\,\mathrm{pp}$. Both approaches are viable since they surpass the current best method. Due to slightly better results, we chose direct forecasting in EAST.

**L2 loss.** Alignment between observed and oracle features is a natural choice in guiding anticipative behaviour. However, Table 9 b) shows that adding an $L2$ loss between oracle and predicted features lowers accuracy by an average of $0.3\,\mathrm{pp}$. We noticed that the $L2$ loss minimizes at the start of training. Our hypothesis is that strong feature alignment neglects some important temporal patterns.

**Shared vs separate classifiers.** Table 9 c) shows the contribution of the shared classification head. The shared classifier slightly outperforms two independent classifiers by an average of $0.3\,\mathrm{pp}$, with a maximum improvement of $0.7\,\mathrm{pp}$ at $\rho = 0.5$. Parameter sharing enforces a consistent decision boundary between observed and predicted features. It also reduces the overall model complexity.

**Wall clock time.** We compare the time duration of one training epoch between EAST and TemPr. We use the same $4 \times$ A6000 server, and turn off unnecessary CPU-GPU communication, such as loss logging. Mean measurements on UCF-101 are 80 seconds for EAST and 173 seconds for TemPr. EAST without masking trains an epoch in 180 seconds, highlighting the token masking efficiency. During inference, EAST processes a video clip in $12\,\mathrm{ms}$, whereas TemPr executes in $78\,\mathrm{ms}$.

**One vs per $\rho$ model.** Figure 2 compares EAST with 9 models that specialize in a single observation ratio. Although specialized models mostly perform better at their respective training-time observation ratio, they fail in most other setups. The results indicate that training with EAST yields

Table 9: Validation of the decoder, loss choice, and classification head on SSv2 against top-1 accuracy across different $\rho$.

a) Direct decoder $\mathcal{D}_{\text{dir}}$ consistently outperforms autoregressive decoder $\mathcal{D}_{\text{ar}}$.

| $\mathcal{D}$ | Observation ratio $\rho$ | | | |
|---|---|---|---|---|
| | 0.1 | 0.3 | 0.5 | 0.7 |
| $\mathcal{D}_{\text{ar}}$ | 25.0 | 34.2 | 48.2 | 58.8 |
| $\mathcal{D}_{\text{dir}}$ | 25.6 | 34.5 | 49.0 | 59.4 |

b) The inclusion of an $\mathcal{L}_2$ loss in EAST yields no further performance gains.

| $\mathcal{L}_2$ | Observation ratio $\rho$ | | | |
|---|---|---|---|---|
| | 0.1 | 0.3 | 0.5 | 0.7 |
| ✗ | 25.6 | 34.5 | 49.0 | 59.4 |
| ✓ | 25.7 | 34.4 | 48.2 | 59.1 |

c) The shared classifer $h$ slightly outperforms separate classification heads.

| cls $h$ | Observation ratio $\rho$ | | | |
|---|---|---|---|---|
| | 0.1 | 0.3 | 0.5 | 0.7 |
| shared | 25.6 | 34.5 | 49.0 | 59.4 |
| separated | 25.7 | 34.3 | 48.3 | 59.2 |

sufficient capacity to learn discriminative cues across different observation ratios, which plays a crucial role in improving model performance. See supplement for more insights.

**Qualitative examples.** Figure 3 shows qualitative comparison between VideoMAE and EAST ViT-B models. We show model outputs at different observation ratios $\rho \in \{0.1, 0.3, 0.5, 0.7, 0.9\}$. The examples show that VideoMAE outputs incorrect classification at small $\rho$, but its outputs are correct at higher $\rho$. In contrast, EAST demonstrates the ability to identify the correct class starting from the lowest observation ratio $\rho = 0.1$. The examples highlight EAST's ability to extract discriminative cues given a portion of the action. More examples can be seen in the Appendix, Figure 4.

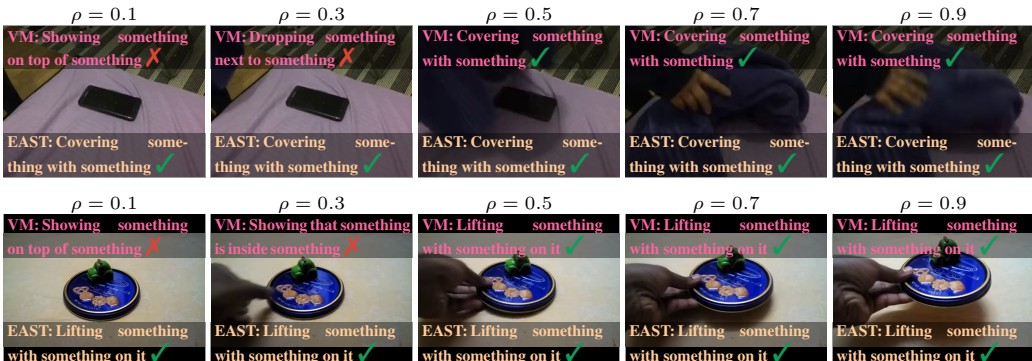

Figure 3: Examples from the SSv2 dataset. We show the last frame within 5 observation ratios. At the specified observation ratio $\rho$, ✗ and ✓ denote false and true model predictions, respectively. EAST makes accurate early predictions, while VideoMAE (VM) requires a larger $\rho$.

## 5 CONCLUSION

Early action prediction is essential for timely decision making in safety-critical domains. This work identified the main components of a successful early action prediction system. We introduced a novel training framework that samples observation ratios in order to adapt the model to variable context length. Unlike the previous best method, this strategy enables training a single model that requires 9× less compute and excels across all observation ratios. We further improved our baseline model by jointly training classification from forecasted and oracle features. Finally, we have proposed a training optimization that removes the visually repetitive half of the inputs, thus halving the training memory. Our results demonstrate that training can be significantly simplified and still outperform the previous state-of-the-art on SSv2, NTU60 and UCF101 using more affordable hardware. Future research directions include unsupervised training and finding a unified method for both action anticipation and early action prediction.

## 6 ACKNOWLEDGMENTS

This work has been supported by Croatian Recovery and Resilience Fund - NextGenerationEU (grant C1.4 R5-I2.01.0001) and the Advanced computing service provided by the University of Zagreb University Computing Centre - SRCE. We thank Ivan Grubišić for carefully reading the manuscript and providing constructive suggestions and insightful comments. We would like to express our deep gratitude to our beloved late professor, Siniša Šegvić. As our mentor, he offered invaluable insight, direction and critical feedback. His intellectual leadership provided a foundation for this work, shaping its development and refinement. We remain sincerely grateful not only for his scientific vision, but also for his kindness, support and lasting influence on both this research and our professional growth.

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

## A    APPENDIX

This appendix is organized as follows. We begin by discussing the method limitations. Next, we validate the decoder depth and measure the sensitivity to training with different random seeds. Furthermore, we demonstrate the effectiveness of the proposed framework using a different backbone. Additionally, we present comprehensive results of our method across all observation ratio values $\rho$. Most of these results are presented in the main paper. However, due to limited space, the main paper does not contain results under all observations ratios. Finally, we present additional analysis of the baseline $EAST_{\mathcal{E}}$ on UCF101 dataset, along with additional qualitative examples.

### A.1    LIMITATIONS

While EAST improves training efficiency and predictive performance, several limitations remain. Since the proposed token masking benefits training more than inference, the inference speed of ViT encoders limits real-time applications at today's hardware. Although we moved the needle towards practical use cases by training a single model agnostic to observation ratios, the model still requires a GPU to operate near real time. Moreover, our encoder does not perform causal inference, which necessitates sliding-window inference over the temporal dimension. This introduces two challenges: i) the minimum decision latency is bounded by the window length $T$, and ii) it prevents streaming inference, which would better capture natural temporal progression and long-term context. Note that evaluating streaming approaches is currently infeasible due to short video duration in existing early action prediction benchmarks.

### A.2    ABLATION ON DECODER DEPTH

Table 10 validates the number of transformer blocks in the decoder $\mathcal{D}$. We evaluate depths of 1, 4 and 12 blocks. The experiments show that decoder depth is an important design choice. The decoder with 4 layers consistently achieves the best accuracy, improving by $0.1$ pp over both 1 and 12 layers on the SSv2 dataset. On the UCF101 dataset, the advantage is more pronounced, with improvements of $0.8$ pp over 1 layer and $1.3$ pp over 12 layers. A 4 layer decoder is expressive enough to transform the encoded features into accurate predictions, yet not too complex to avoid potential overfitting.

Table 10: Top-1 accuracy of  EAST on SSv2 and UCF101 for three different decoder depths: 1, 4 and 12. The best results are in **bold**.

| Dataset | Depth | Observation ratio $\rho$ | | | | | | | | | |
|---|---|---|---|---|---|---|---|---|---|---|---|
| | | 0.1 | 0.2 | 0.3 | 0.4 | 0.5 | 0.6 | 0.7 | 0.8 | 0.9 | avg |
| | 1 | 25.4 | 29.9 | 34.1 | 41.2 | 48.6 | 55.1 | 59.5 | **63.0** | 64.0 | 46.8 |
| SSv2 | 4 | **25.6** | **30.1** | **34.5** | **41.6** | **49.0** | **55.2** | 59.4 | **63.0** | 64.0 | **46.9** |
| | 12 | **25.6** | 30.0 | 34.2 | 41.4 | 48.7 | 55.1 | **59.6** | 62.7 | **64.1** | 46.8 |
| | 1 | 90.5 | 92.4 | 93.1 | 94.3 | 94.7 | 95.0 | 95.6 | 95.4 | 95.5 | 94.1 |
| UCF101 | 4 | **91.3** | **93.2** | **93.8** | **94.7** | **95.5** | **96.1** | **96.4** | **96.5** | **96.5** | **94.9** |
| | 12 | 90.3 | 92.0 | 92.8 | 93.6 | 94.3 | 94.6 | 94.9 | 94.9 | 94.9 | 93.6 |

### A.3    ROBUSTNESS OF EAST TO RANDOM SEED

Our main results in Section 4 use a fixed seed within a single training run. We have found this to be common practice in prior work. Nonetheless, we demonstrate that our method is not sensitive to random seed selection. We perform additional training runs on Something-Something v2 with two more random seeds. Table 11 reports $mean_{\pm std}$ over three runs to confirm low sensitivity to randomness in training.

### A.4    VALIDATION OF EAST USING A MOVINET BACKBONE

The first row in Table 12 shows the accuracy of an encoder-only model with a MoViNet backbone. The second row in Table 12 shows that training encoder-only $EAST_{\mathcal{E}}$ with MoViNet backbone

Table 11: Top-1 accuracy (%) of EAST over all observation ratios for the SSv2, reported as the $mean_{\pm std}$ over three random seeds.

| Dataset | Observation ratio $\rho$ | | | | | | | | |
|---|---|---|---|---|---|---|---|---|---|
| | 0.1 | 0.2 | 0.3 | 0.4 | 0.5 | 0.6 | 0.7 | 0.8 | 0.9 |
| SSv2 | $25.5_{\pm 0.2}$ | $29.8_{\pm 0.3}$ | $34.2_{\pm 0.3}$ | $41.1_{\pm 0.4}$ | $48.6_{\pm 0.4}$ | $55.0_{\pm 0.2}$ | $59.3_{\pm 0.1}$ | $62.7_{\pm 0.3}$ | $63.7_{\pm 0.3}$ |

Table 12: Contributions of the proposed losses and modules to UCF101 validation accuracy with MoViNet encoder. $\mathcal{D}$ denotes the choice of the decoder, id denotes an identity mapping (encoder-only), whereas ✓uses the proposed decoder $\mathcal{D}$.

| $\mathcal{L}^{\text{oracle}}$ | $\mathcal{L}^{\text{pred}}$ | $\mathcal{D}$ | $\rho{=}0.1$ | $\rho{=}0.2$ | $\rho{=}0.3$ | $\rho{=}0.4$ | $\rho{=}0.5$ | $\rho{=}0.6$ | $\rho{=}0.7$ | $\rho{=}0.8$ | $\rho{=}0.9$ | avg |
|---|---|---|---|---|---|---|---|---|---|---|---|---|
| | ✓ | id | 88.3 | 90.4 | 91.2 | 91.4 | 92.1 | 92.5 | 92.7 | 92.8 | 92.9 | 91.6 |
| ✓ | ✓ | id | 88.7 | 90.7 | 91.8 | 92.6 | 93.8 | 94.4 | 94.3 | 94.4 | 94.5 | 92.8 |
| ✓ | ✓ | ✓ | 91.3 | 93.2 | 93.8 | 94.7 | 95.5 | 96.1 | 96.4 | 96.5 | 96.5 | 94.9 |

using both $\mathcal{L}^{\text{pred}}$ and $\mathcal{L}^{\text{oracle}}$ gains additional 1.2 pp. Training an encoder-decoder model using $\mathcal{L}^{\text{pred}}$ and $\mathcal{L}^{\text{oracle}}$ (*cf.* EAST$_{\text{MoViNet}}$ from Table 4) further improves the average accuracy by 2.1 pp. The results highlight the benefits of training using the proposed compound loss in both cases, regardless of the backbone. Note that training with MoViNet limits the batch size to 32 since token masking is not applicable. In comparison, ViT-B/16 supports a larger batch size of 128 on the same GPUs.

## A.5 EAST RESULTS PER $\rho$ ON NTU60, SSSUB21 AND EK-100

Table 13 reports top-1 accuracy EAST obtains at each observation ratio on the NTU60, SSsub21 and EK-100 datasets. We provide a detailed performance comparison across the full range of evaluated observation ratios.

## A.6 EAST ABLATION RESULTS ACROSS ALL OBSERVATION RATIOS

Table 14 shows the accuracy of EAST for every observation ratio $\rho$ when using different decoders. Direct decoder $\mathcal{D}_{\text{dir}}$ shows consistent improvement for all observation ratios in comparisons to autoregressive decoder $\mathcal{D}_{\text{ar}}$. On average, the direct decoder yields a 0.3 pp improvement in accuracy.

Table 15 shows the accuracy EAST obtains when training with $\mathcal{L}_2$ loss in conjunction with the proposed classification losses. We notice only marginal accuracy increase of 0.1 pp for $\rho = 0.1$. At other observation ratios, there is no benefit of using the $\mathcal{L}_2$ loss. On average, using only the classification losses improves accuracy by 0.3 pp.

Table 16 shows the accuracy of EAST for every observation ratio $\rho$ when we use one classification head and when we use separate classification heads. We notice minimal gain in accuracy of 0.1 pp for observation ratio $\rho = 0.1$. Using a single classification head clearly improves the average accuracy by 0.3 pp.

## A.7 PERFORMANCE OF EAST VS SINGLE MODEL FOR SINGLE $\rho$

Table 17 shows that training one model for each $\rho$ can match or occasionally surpass our performance at its own observation ratio $\rho$. However, its accuracy deteriorates noticeably when applied to other ratios, with the decline becoming more severe as the evaluation ratio diverges from the training ratio. In contrast, EAST maintains consistently strong performance across all observation ratios, demonstrating greater robustness to changes in $\rho$.

Table 13: Extended Tables 1, 3 and 5 from the main paper. Top-1 accuracy (%) of EAST over all observation ratios for the NTU60, SSsub21 and EK-100 datasets.

| Dataset | Observation ratio $\rho$ | | | | | | | | |
|---|---|---|---|---|---|---|---|---|---|
| | 0.1 | 0.2 | 0.3 | 0.4 | 0.5 | 0.6 | 0.7 | 0.8 | 0.9 |
| NTU60 | 31.2 | 49.6 | 69.4 | 81.3 | 86.2 | 87.6 | 87.9 | 88.0 | 87.9 |
| SSsub21 | 40.8 | 44.7 | 51.2 | 59.2 | 66.4 | 72.0 | 75.8 | 78.3 | 79.3 |
| EK-100$_{\text{All Action}}$ | 20.4 | 23.3 | 25.4 | 27.0 | 28.1 | 29.0 | 29.7 | 29.9 | 29.6 |
| EK-100$_{\text{All Noun}}$ | 31.1 | 34.0 | 35.5 | 37.3 | 38.2 | 39.3 | 39.9 | 40.5 | 40.1 |
| EK-100$_{\text{All Verb}}$ | 47.2 | 52.1 | 55.0 | 56.7 | 58.4 | 59.1 | 59.5 | 59.8 | 58.4 |

Table 14: Extended Table 9 a) from the main paper. Top-1 accuracy of EAST on SSv2 for each $\rho$. Direct decoder $\mathcal{D}_{\text{dir}}$ consistently outperforms autoregressive decoder $\mathcal{D}_{\text{ar}}$.

| $\mathcal{D}$ | Observation ratio $\rho$ | | | | | | | | | |
|---|---|---|---|---|---|---|---|---|---|---|
| | 0.1 | 0.2 | 0.3 | 0.4 | 0.5 | 0.6 | 0.7 | 0.8 | 0.9 | avg |
| $\mathcal{D}_{\text{dir}}$ | **25.6** | **30.1** | **34.5** | **41.6** | **49.0** | **55.2** | **59.4** | **63.0** | **64.0** | **46.9** |
| $\mathcal{D}_{\text{ar}}$ | 25.0 | 29.5 | 34.2 | 40.9 | 48.1 | 54.5 | 58.8 | 62.4 | 63.3 | 46.3 |

Table 15: Extended Table 9 b) from the main paper. Top-1 accuracy of EAST on SSv2 over all reported observation ratios when using $\mathcal{L}_2$ in addition to classification loss.

| $\mathcal{L}_2$ | Observation ratio $\rho$ | | | | | | | | | |
|---|---|---|---|---|---|---|---|---|---|---|
| | 0.1 | 0.2 | 0.3 | 0.4 | 0.5 | 0.6 | 0.7 | 0.8 | 0.9 | avg |
| ✗ | 25.6 | **30.1** | **34.5** | **41.6** | **49.0** | **55.2** | **59.4** | **63.0** | **64.0** | **46.9** |
| ✓ | **25.7** | 29.8 | 34.4 | 40.9 | 48.2 | 54.7 | 59.1 | 62.6 | 63.7 | 46.6 |

Table 16: Extended Table 9 c) from the main paper. Top-1 accuracy of EAST on SSv2 over all reported observation ratios when using shared classification head vs separate classification heads for different set of features.

| # cls $h$ | Observation ratio $\rho$ | | | | | | | | |
|---|---|---|---|---|---|---|---|---|---|
| | 0.1 | 0.2 | 0.3 | 0.4 | 0.5 | 0.6 | 0.7 | 0.8 | 0.9 |
| 1 | 25.6 | **30.1** | **34.5** | **41.6** | **49.0** | **55.2** | **59.4** | **63.0** | **64.0** |
| 2 | **25.7** | 30.0 | 34.3 | 41.0 | 48.3 | 54.7 | 59.2 | 62.5 | 63.5 |

Table 17: Numerical values for the Figure 2 in the main paper. Top-1 accuracy on SSv2 over all observation ratios when we train one model for each $\rho$ vs EAST. Results for the matching training $\rho$ are shown in **bold**.

| model | Observation ratio $\rho$ | | | | | | | | |
|---|---|---|---|---|---|---|---|---|---|
| | 0.1 | 0.2 | 0.3 | 0.4 | 0.5 | 0.6 | 0.7 | 0.8 | 0.9 |
| EAST | 25.6 | 30.1 | 34.5 | 41.6 | 49.0 | 55.2 | 59.4 | 63.0 | 64.0 |
| $\rho = 0.1$ | **27.1** | 27.4 | 29.2 | 29.2 | 27.1 | 27.1 | 24.3 | 22.0 | 20.1 |
| $\rho = 0.2$ | 24.7 | **32.5** | 33.0 | 35.9 | 37.8 | 37.8 | 36.2 | 33.5 | 31.0 |
| $\rho = 0.3$ | 22.2 | 30.2 | **34.9** | 39.6 | 42.7 | 44.1 | 43.1 | 42.6 | 39.0 |
| $\rho = 0.4$ | 21.0 | 28.0 | 34.1 | **41.1** | 46.5 | 49.9 | 50.9 | 50.1 | 48.4 |
| $\rho = 0.5$ | 18.4 | 25.4 | 31.6 | 40.3 | **50.1** | 53.8 | 56.9 | 55.4 | 55.0 |
| $\rho = 0.6$ | 15.7 | 22.0 | 27.8 | 44.6 | 47.0 | **54.5** | 58.5 | 58.5 | 60.1 |
| $\rho = 0.7$ | 13.0 | 19.0 | 25.3 | 42.3 | 44.7 | 53.9 | **59.6** | 62.7 | 62.7 |
| $\rho = 0.8$ | 10.7 | 16.1 | 19.8 | 31.0 | 41.7 | 51.6 | 58.8 | **63.7** | 64.7 |
| $\rho = 0.9$ | 9.7 | 15.9 | 19.1 | 28.6 | 38.8 | 49.3 | 57.3 | 64.0 | **64.9** |

Table 18: Extended Table 7 from the main paper. Top-1 accuracy of EAST on NTU60 over all reported observation ratios for different masking setups. $\mathcal{M}^{\mathrm{d}}$, $\mathcal{M}^{\mathrm{MAR}}$ and $\mathcal{M}^{\mathrm{rand}}$ denote difference masking, Running Cell masking and random masking, respectively. $k$ denotes percentage of masked tokens. w/o denotes no masking of video frames. TFLOP denotes the number of floating point operations. Peak mem. denotes the maximum amount of GPU memory allocated at any point during execution.

| | Observation ratio $\rho$ | | | | | | | | | | | |
| Masking | 0.1 | 0.2 | 0.3 | 0.4 | 0.5 | 0.6 | 0.7 | 0.8 | 0.9 | avg | TFLOP | peak mem. (GB) |
|---|---|---|---|---|---|---|---|---|---|---|---|---|
| $\mathcal{M}^{\mathrm{rand}}_{k=0.75}$ | 23.3 | 36.5 | 56.3 | 70.3 | 76.5 | 77.9 | 78.4 | 78.5 | 78.5 | 64.0 | 0.24 | 10.4 |
| $\mathcal{M}^{\mathrm{MAR}}_{k=0.75}$ | 27.3 | 40.9 | 59.4 | 72.2 | 78.0 | 79.3 | 79.7 | 79.8 | 79.8 | 66.3 | | |
| $\mathcal{M}^{\mathrm{d}}_{k=0.75}$ | 28.4 | 46.4 | 65.9 | 78.3 | 84.0 | 85.6 | 86.1 | 86.1 | 86.1 | 71.9 | | |
| $\mathcal{M}^{\mathrm{rand}}_{k=0.5}$ | 28.6 | 45.7 | 66.1 | 78.3 | 83.5 | 84.6 | 85.0 | 85.1 | 85.0 | 71.3 | 0.5 | 19.2 |
| $\mathcal{M}^{\mathrm{MAR}}_{k=0.5}$ | 31.3 | 47.8 | 67.1 | 79.0 | 83.8 | 85.4 | 85.8 | 85.8 | 85.8 | 72.4 | | |
| $\mathcal{M}^{\mathrm{d}}_{k=0.5}$ | 31.2 | 49.6 | 69.4 | 81.3 | 86.2 | 87.6 | 87.9 | 88.0 | 87.9 | 74.3 | | |
| $\mathcal{M}^{\mathrm{rand}}_{k=0.25}$ | 31.1 | 48.3 | 67.9 | 79.8 | 85.1 | 86.8 | 87.2 | 87.3 | 87.3 | 73.4 | 0.8 | 27.9 |
| $\mathcal{M}^{\mathrm{MAR}}_{k=0.25}$ | 31.5 | 48.1 | 67.8 | 79.8 | 84.9 | 86.6 | 86.9 | 86.9 | 86.9 | 73.3 | | |
| $\mathcal{M}^{\mathrm{d}}_{k=0.25}$ | 31.9 | 51.0 | 70.8 | 81.9 | 86.7 | 88.3 | 88.6 | 88.7 | 88.6 | 75.2 | | |
| w/o mask | 32.6 | 50.7 | 70.3 | 82.1 | 86.9 | 88.1 | 88.4 | 88.5 | 88.5 | 75.1 | 1.1 | 36.7 |

## A.8 RESULTS OF DIFFERENT MASKING SETUPS FOR EACH $\rho$

Table 18 shows that our chosen masking strategy $\mathcal{M}^{\mathrm{d}}_{k=0.5}$ demonstrates a clear and consistent advantage across all observation ratios $\rho$. By selectively retaining the most informative tokens, it effectively balances predictive accuracy and computational cost. This approach serves as an optimal middle ground, delivering strong performance while avoiding the excessive resource demands.

## A.9 ADDITIONAL ANALYSIS OF THE OVERLOOKED BASELINE $\mathrm{EAST}_{\mathcal{E}}$ ON UCF101

Table 19 presents additional evaluation of $\mathrm{EAST}_{\mathcal{E}}$ on UCF101. The results show that VideoMAE reaches competitive accuracy for higher $\rho$. This trend is present in Table 6 on SSv2. Additional analysis of Table 17 clarifies why VideoMAE achieves competitive results at higher observation ratios. The table contains models specifically trained at one observation ratio $\rho$. We can observe that these models excel around $\rho$ they trained at, whereas their accuracy drops when moving away from that specific $\rho$. Since VideoMAE is a model specialized for $\rho = 1.0$, we observe the same behavior: VideoMAE performs best around observation ratio it is optimized for, i.e. at higher $\rho$ values.

Table 19: Top-1 UCF101 accuracy over all observation ratios. VideoMAE denotes pre-trained ViT-B/16 model performance finetuned for action classification. $\mathrm{EAST}_{\mathcal{E}}$ trains ViT-B/16 with our proposed sampling without a decoder.

| | Observation ratio $\rho$ | | | | | | | | |
| Method | 0.1 | 0.2 | 0.3 | 0.4 | 0.5 | 0.6 | 0.7 | 0.8 | 0.9 |
|---|---|---|---|---|---|---|---|---|---|
| VideoMAE | 67.6 | 77.3 | 80.5 | 82.2 | 84.5 | 84.5 | 84.9 | 85.0 | 85.0 |
| $\mathrm{EAST}_{\mathcal{E}}$ | 79.5 | 82.4 | 83.7 | 84.3 | 84.6 | 84.7 | 85.0 | 85.0 | 85.0 |

## A.10 ADDITIONAL QUALITATIVE EXAMPLES

Additional qualitative examples are presented in Figure 4, highlighting the ability of EAST to make correct predictions starting from $\rho = 0.1$. We observe that for examples such as Lifting something with something on it (row 3) and Spinning something that stops (row 4), the VideoMAE is able to make the correct decision relatively early. However, for examples like Opening something (row 1) and Dropping something into something (row 2), VideoMAE needs to observe roughly $70 - 90\%$ of the video to make reliable classification. In some instances, like Closing something (row 5), VideoMAE still fails

to predict the correct class even after observing $90\%$ of the video. On the other hand, EAST can make the correct decision even for the smallest observation ratio $\rho = 0.1$.

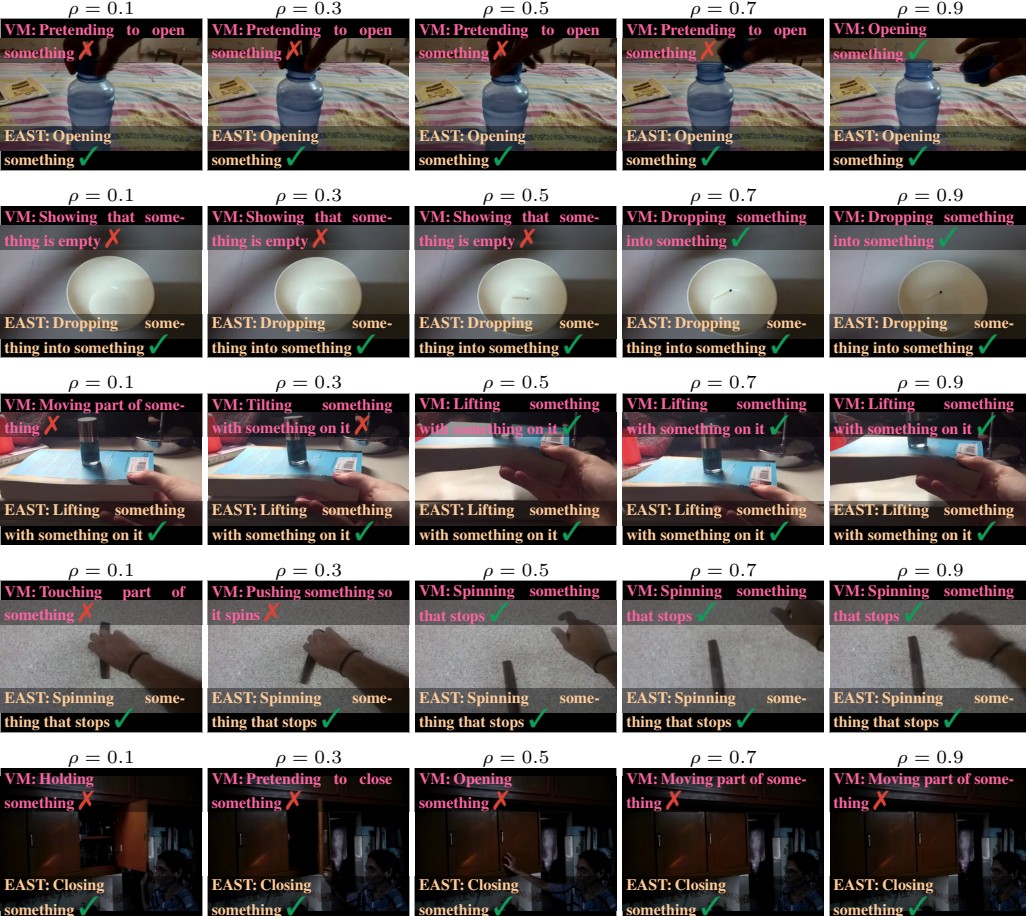

Figure 4: Qualitative analysis on SSv2. We show the last frame within 5 observation ratios. At the specified observation ratio $\rho$, ✗ and ✓ denote false and true model predictions, respectively.

