# OpenReview forum: "EAST: Early Action Prediction Sampling Strategy with Token Masking"
_ICLR.cc/2026/Conference — ICLR 2026 Poster_

### Official Review · Reviewer_mSTD · 2025-10-26

**Soundness:** 3
**Presentation:** 3
**Contribution:** 3
**Rating:** 6
**Confidence:** 4

**Summary:**

This paper reseaches on action early recogntion (or action prediction, coined by ealier work) in videos. Insteading of pursuing complicated techniques, this paper actually advocates and demonstrates that simple model architecture and training schema would be sufficient to have good results, as well as be sufficient to accomendate a varitety of inference requirements (e.g., observation ratios in this use case). Strong empirical results support the claim.

**Strengths:**

A. Very impressive evaluation results - consistently improving on popular benchmarks by notable margin.

B. The overall approach bears a good balance of simplicity - the model architecture being transformers (i.e., a very general and prior-free model family), the optimization strategy (i.e., the training loss is a standard cross-entropy loss), the design for being agnostic against observation ratios with one unified approach, and lastly the hyper-parameters also being straightforward and easy to implement.

**Weaknesses:**

A. This review has strong concern on the assumptions incoporated and implemented in the token masking module, which is proposed for the purpose of the effienct model design - repetitive frame masking. This strategy, though indeed reduces FLOPS and speeds up training/inference, pre-assumes that similar tokens should be ignored and largely different tokens in feature space should be attended to more, which is arguable when it comes to a boarder scope of action early recongtion task. Imagine fast action video clips such as "world chase tag" competition where the frames are mostly motion intense, have much less idle motions and even the idle period is quite important. It is very unlikely that the 50% frame masking strategy would hold all essential information. Even more, let's assume 50% masking rate is appropriate, the masking decision comes from the token-wise similarity captured by feature space difference. This strategy has inherent flaws, such as big background motion will be reckoned as more valuable compared to small actor related motion under current fashsion. From the experimental results, it seems that using full frame (or token) still bring the best results, which makes the masking less critical. To sum up, this reviewer appreciate the efforts on making slim model, but disagree on doing so depending on any prior assumptions, such as feature space difference.

B. Can the authors further clarify the protocol difference between the proposed EAST method and the most recent work, e.g., Tempr? This reviewer found an evaluation tip, i.e., multi-crop inference, a training tip, i.e., data augmentaion, and a scaled learning rate schedule, in the paper and wonders if the comparison method follow the same procedure. Sometimes, training and evaluation details can lead to big difference. It would be nice to keep the protocol the same on both side. This reviewer conjure that difference in optimization could has strong impact on final results.

C. How is the VideoMAE trained in Table 5? Is it just using the observed frames at a certain ratio to predict the action label? Is it also a observation ratio agnostic model? Why the results indicate the impact of "sampling strategy"? What is the loss and input here?

D. This reviewer is particularly interested in Table 5 and would like to observe more on the performance gain when being compared to VideoMAE from other dataset. Actually, VideoMAE is more comparable to EAST than other early action recognition work (e.g., Tempr) in terms of model architecture similarity and loss objectives. It will be a big help if authors can add such comparison to all benchmarks and spot any interesting patterns. One thing noticed by this reviewer from the current Table 5 is that the 0.9 observation ratio accuracy from EAST_{epson} is deteriorated, i.e., 60.7 vs. 63.4. And this makes reviewer wonder if current VideoMAE is biased on later frames (i.e., focus on 80th - 90th percent frames of a video clip). If so, it seems logically reasonable that the proposed training strategy depicted in figure 1 resolves such bias to some extent, or say learn a more generalized classifier across time - The L^{pred} will object when L^{oracle} optimize later frames too much. The learning dynamic here can be a very interesting investigation. Maybe plotting the L^{oracle} of 80th-90th percent frames vs L^{pred} on the validation set during training would be insightful, which also fits the taste of ICLR (more machine learning analysis).

**Questions:**

N/A

---

> ### Author Response · Authors · 2025-11-23
> **Response to Reviewer mSTD**
>
> We thank the reviewer for constructive criticism! We are glad that the reviewer recognized the efficiency benefits proposed within EAST. The initial review compelled us to make additional experiments that could bring more interesting insights to readers.
> We address the weaknesses in order of original comments:
>
> >W.A assumptions incorporated and implemented in the token masking
>
> Table 6 in the main paper shows that masking fewer input tokens improves early action prediction accuracy. However, this comes at a significant computational cost. This cost motivated our research on memory efficient training. Table 6 demonstrates that the proposed difference masking consistently improves over random masking $\mathcal{M}_k^{\text{rand}}$ within the same computational budget. "*disagree on doing so depending on any prior assumptions*": Our new experiments in this rebuttal include Running Cell masking from MAR[^1]. Please see the first table in our response to reviewer 9qPd. This masking strategy selects highly diverse cubes to capture information from all parts of the video. This masking removes 50% of tokens over a non-random regular grid. The results show that MAR masking $\mathcal{M}_k^{\text{MAR}}$ improves over random masking, but under performs compared to $\mathcal{M}_k^{\text{d}}$. Masking with $\mathcal{M}_k^{\text{MAR}}$ does not make any assumptions on the input data, but falls short compared to EAST. On standard early action prediction datasets we do not find that $\mathcal{M}_k^{\text{d}}$ degrades performance achieved using $\mathcal{M}_k^{\text{rand}}$.
>
> >W.B the protocol difference
>
> Thank you for the careful analysis of the evaluation protocol. Our implementation is based on a VideoMAE and uses the same hyperparameters as original work, unless stated otherwise. We also trained with the MoViNet backbone using the same codebase, for fairness. We fork [TemPR](https://github.com/alexandrosstergiou/progressive-action-prediction) when evaluating computational complexity on common hardware. For fairness, our timing and memory measurements include the computational burden of multi crop evaluation.
> Both EAST and TemPr have standard data augmentation during training and validation as center crop and normalization, since it is a typical protocol for any video model. Note that TemPr model architecture includes 4 temporal samples during both training and evaluation. EAST includes 2 temporal samples during evaluation, and we find that additional samples do  not provide any further improvement in accuracy.
>
> >W.C How is the VideoMAE trained in Table 5
>
> Thank you for highlighting Table 5 since it illustrates that action recognition models fall short when applied to early action recognition. Both rows use the same ViT-B architecture and are trained with common hyperparameters. VideoMAE entry show results obtained by evaluating the public [action recognition checkpoint](https://github.com/MCG-NJU/VideoMAE/blob/main/MODEL_ZOO.md#:~:text=92.0-,VideoMAE,-no) to observed data defined by EAST sampling. ${\text{EAST}}_{\mathcal{E}}$ entry trains ViT-B using the proposed sampling strategy and does not use a decoder.
>
> This model has a single classification head trained with ${\mathcal{L}}_{\text{pred}}$. Importantly, the models are completely agnostic to the testing observation ratio. Since VideoMAE observes full video clips in training, it is natural that reducing the observation ratio degrades it's performance.
>
> >W.D performance gain when being compared to VideoMAE
>
> We find this comment very insightful and are currently running experiments to investigate this.
>
> [EDIT 2025-12-01: The experiment runs are complete. We present the results in a comment below.]
>
>
> [^1]: Qing, Zhiwu, et al. "Mar: Masked autoencoders for efficient action recognition." IEEE Transactions on Multimedia 26 (2023): 218-233.

---

> > ### Comment · Reviewer_mSTD · 2025-11-25
> >
> > Hi,
> >
> > >by evaluating the public action recognition checkpoint to observed data defined by EAST sampling
> >
> > Thanks for the clarification. This study compares the evaluation results of 1) VideoMAE pre-trained on a set-up that it was not originally trained for (e.g., assuming the sampling strategy used in EAST is not the sampling strategy used for VideoMAE training) v.s the EAST model (same VideoMAE backbone plus a compound loss) trained and evaluated on the same sampling strategy. Then, this study becomes less reasonable to this reviewer, as there are two variables: the compound loss + sampling strategy.  Shouldn't a reasonable comparison be
> >
> > - a pre-trained VideoMAE evaluated on new sampling strategy vs.
> >
> > - a same VideoMAE but trained/evaluated on the new sampling strategy
> >
> > (?)

---

> ### Author Response · Authors · 2025-11-25
> **Response to Reviewer mSTD**
>
> Thank you for the follow-up. We apologize if our previous explanation made it sound otherwise, but in Table 5 we actually evaluate exactly the two cases you described.
>
> - First row: A pre-trained VideoMAE fine-tuned on SSv2 and evaluated under the standard early action prediction evaluation protocol ($\rho$ from 0.1 to 0.9) using our sampling strategy.
>
> - Second row: $EAST_{\varepsilon}$, which is  VideoMAE fine-tuned on SSv2 with our sampling strategy and evaluated under the same standard early action prediction evaluation protocol ($\rho$ from 0.1 to 0.9) using our sampling strategy.
> Please note that $EAST_{\varepsilon}$ does not use the compound loss and is therefore different from our final EAST model.
>
> We will make sure to revise the paper and Table 5 to clarify this more explicitly.

---

> > ### Comment · Reviewer_mSTD · 2025-11-26
> >
> > Thanks for the further clarification.
> >
> >
> > This is an insightful information (even more important than other parts of the paper such as model or loss designs, as least to this reviewer), as it reveals that a different task (i.e., early action recognition) could really use a different information preparation strategy (i.e., EAST sampling). This is an interesting investigation on the problem from bottom to up, showing a deeper thoughts on this problem.

---

> > > ### Author Response · Authors · 2025-12-01
> > > **Response to Reviewer mSTD**
> > >
> > > As promised during the initial rebuttal, we performed the additional experiment to supplement our answer to W4.
> > >
> > > Since VideoMAE is pre-trained on UCF101, we run additional experiment on UCF101. The table bellow presents the results.
> > > | $\rho$                      |  0.1 |  0.2 |  0.3 |  0.4 |  0.5 |  0.6 |  0.7 |  0.8 |  0.9 |
> > > | --------------------------- | ---: | ---: | ---: | ---: | ---: | ---: | ---: | ---: | ---: |
> > > | VideoMAE                    | 67.6 | 77.3 | 80.5 | 82.2 | 84.5 | 84.5 | 84.9 | 85.0 | 85.0 |
> > > | $\text{EAST}_{\varepsilon}$ | 79.5 | 82.4 | 83.7 | 84.3 | 84.6 | 84.7 | 85.0 | 85.0 | 85.0 |
> > >
> > > The results still show that VideoMAE has competitive accuracy for higher $\rho$.
> > >
> > > Detailed analysis of Table 17, which we include in Appendix A.8, gives some more clarity to why VideoMAE outperforms $\text{EAST}_{\varepsilon}$ at higher observation ratios. The table contains models specifically trained at one observation ratio ρ. We can observe that each model excels around ρ that it was trained for, whereas its accuracy drops when moving away from that specific ρ. Since VideoMAE is a model specialized for ρ=1.0, we observe the same behavior: VideoMAE performs best around observation ratio it is optimized for, which are higher rho values.
> > >
> > > We will include this discussion and the additional results in the final manuscript.

---

### Official Review · Reviewer_9qPd · 2025-10-27

**Soundness:** 2
**Presentation:** 2
**Contribution:** 1
**Rating:** 2
**Confidence:** 3

**Summary:**

This paper presents EAST, a simple yet effective framework for early action prediction (EAP) that enables a single model to generalize across multiple observation ratios. The method introduces a randomized temporal sampling strategy to jointly train on both observed and unobserved video segments, combined with a token masking technique that removes redundant patches for more efficient training. Extensive experiments on NTU60, Something-Something V2, and UCF101 demonstrate state-of-the-art accuracy while reducing training time and memory by half.

**Strengths:**

S1. Simple yet effective: The proposed sampling and masking strategies are conceptually simple but yield strong performance gains across benchmarks.

S2. In-depth analysis: Extensive ablation studies support each design choice and show consistent improvements.

S3. Comprehensive evaluation: EAST is evaluated on multiple datasets, demonstrating generality and efficiency.

**Weaknesses:**

W1. Weak motivation & significance: The paper does not clearly elaborate which specific limitations in prior EAP methods EAST addresses or why its components are crucial. While the sampling strategy is intuitive and well-justified, the roles of the compound forecasting loss and token masking are underexplained. Moreover, the significance of the tackled problem is not sufficiently emphasized. Clarifying both the motivation and broader impact of the proposed framework, along with the necessity of each component, would substantially strengthen the paper’s contribution.

W2. Outdated baselines: Many comparisons are against older methods; TemPr is the only recent baseline, but its architectural and training differences make fair comparison difficult. Additional controlled analysis on temporally challenging datasets (e.g., Something-Something) would clarify the contribution.

W3. Missing inference speed comparison: Since EAP is latency-sensitive, comparing inference speed with prior works (e.g., TemPr, ERA) would strengthen the claims.

W4. Limited masking comparison: Token masking is only compared with random masking. Including modern masking methods such as VideoMAE [a] or MAR [b] would better validate the approach.

W5. Missing results on Epic-Kitchens 100: Although Epic-Kitchens 100 is a standard benchmark for action anticipation and applicable to EAP (as demonstrated in TemPr), the paper omits it. Including this dataset would better validate generalization in real-world scenarios.

[a] Tong et al., “VideoMAE: Masked Autoencoders are Data-Efficient Learners for Self-Supervised Video Pre-Training,” NeurIPS, 2022. \
[b] Qing et al., “MAR: Masked Autoencoders for Efficient Action Recognition,” Transcations on Multimedia, 2023.

**Questions:**

Please refer to the weakness section above.

---

> ### Author Response · Authors · 2025-11-23
> **Response to Reviewer 9qPd (1/2)**
>
> We thank the reviewer for the motivating review. We address the weaknesses in the same order:
> >W1. Motivation & significance
>
> We find that both token masking and the compound loss are important components in EAST. Token masking is used to substantially improve memory and time efficiency for video analysis. In our case, it reduced training time from 180 seconds to 80 seconds on UCF101 and reduced operations count from 1.1 TFLOP to 0.5 TFLOP. These findings are presented in Table 6 and in the "**Wall clock time**" paragraph. Furthermore, the ablation experiments suggest that the compound loss improves generalization (*cf*. Table 7) and is superior to the commonly used L2 loss (*cf*. Table 8 b). We believe that the early action recognition task is relevant, as also noted by other reviewers. We will improve the discussion regarding the relevance of this research area in the manuscript accordingly.
>
> >W2. Analysis on temporally challenging datasets (e.g., Something-Something)
>
> We believe a misunderstanding occurred, as we perform most of the experiments in the original version of our paper on the Something-Something dataset: Table 2 and most validation experiments evaluate on SSv2. We would appreciate the reviewer's clarification regarding this issue. "*fair comparison*": Our UCF-101 experiment with MoViNet removes the influence of the encoder, since TemPr uses that backbone. Moreover, we would appreciate references to related recent work since we did not find relevant published papers newer than TemPr that were accepted for publication in a journal or conference. If we omitted any, we would gladly correct this.
>
> >W3. Missing inference speed comparison
>
> Thank you for encouraging us to include run-time measurements. Processing a single clip using EAST takes 12ms on the A6000 GPU, while e.g. TemPr takes 78.5ms.
>
> >W4. Limited masking comparison
>
> We use the same random masking strategy as VideoMAE and refer to it as $\mathcal{M}^{\{k=50\}}_{\text{rand}}$ in our paper.
>
> We find that Running Cell masking is very interesting and implement MAR within our codebase [^1]. The results show that it outperforms random masking, but falls short of the proposed difference masking $\mathcal{M}^{\{k=50\}}_{\text{diff}}$. The results are obtained on NTU60 dataset and can be seen in the table below. We will add the results and cite the MAR paper in the revised manuscript accordingly.
>
> | $\rho$                                   |  0.1 |  0.2 |  0.3 |  0.4 |  0.5 |  0.6 |  0.7 |  0.8 |  0.9 |
> | ---------------------------------------- | ---: | ---: | ---: | ---: | ---: | ---: | ---: | ---: | ---: |
> | $\mathcal{M}^{k=50}_{\text{rand}}$ Acc@1 | 28.6 | 45.7 | 66.1 | 78.3 | 83.5 | 84.6 | 85.0 | 85.1 | 85.0 |
> | $\mathcal{M}^{k=50}_{\text{MAR}}$ Acc@1  | 31.3 | 47.8 | 67.1 | 79.0 | 83.8 | 85.4 | 85.8 | 85.8 | 85.8 |
> | $\mathcal{M}^{k=50}_{\text{diff}}$ Acc@1 | 31.2 | 49.6 | 69.4 | 81.3 | 86.2 | 87.6 | 87.9 | 88.0 | 87.9 |
>
>
>
> [^1]: Qing, Zhiwu, et al. "Mar: Masked autoencoders for efficient action recognition." IEEE Transactions on Multimedia 26 (2023): 218-233.

---

> ### Author Response · Authors · 2025-11-23
> **Response to Reviewer 9qPd (2/2)**
>
> >W5. Missing results on Epic-Kitchens 100
>
> We thank the reviewer for suggesting experiments on Epic-Kitchens 100 (EK100).
> We train and evaluate EAST and present the results in this rebuttal.
>
> We are unfortunately unable to comment and reproduce the exact training and evaluation procedure used in TemPr since their repository does not include any scripts for Epic-Kitchens100. It is therefore unclear whether they trained separate models for verbs and nouns, or used a unified setup and what is their evaluation protocol.
> Another source of uncertainty is the difference in backbone since TemPr uses SlowFast for EK100. SlowFast appears to be trained on 32 frames, and TemPr with 54, which may also perhaps make it a more suitable backbone for TemPr.
> However, since the authors did not describe these details nor provided the code, we cannot make definitive statements.
> Also, it would take unreasonable amount of time and resources to run all these alternatives to maybe discover training and evaluation protocol they used.
>
> To the best of our possibilities, we train and evaluate EAST as is on EK100, using two classification heads, one for verb and one for noun, and evaluate using evaluation scripts provided by the official EK100 repository.
> The results can be seen in the tables below.
> The results suggest that EAST outperforms TemPr at low observation ratios but is limited by the ViT encoder accuracy at high observation ratios.
> TemPr uses the SlowFast backbone that achieves top1 action recognition accuracy of 38.5%, 50.0% and 65.6% on *all action*, *all noun* and *all verb*, respectively.
> This outperforms VideoMAE ViT-B that achieves  33.7%, 42.7% and 65.1% top1 accuracy on *all action*, *all noun* and *all verb*, respectively.
>
> This further strengthens our contributions, since EAST significantly outperforms TemPr on low-observation ratios, which are of the most interest for early action prediction. Since TemPr uses a stronger backbone, it is expected that it obtains better results at larger observation ratios.
>
>
>  ### EK100  all action
>
> | $\rho$      |  0.1 |  0.2 |  0.3 |  0.4 |  0.5 |  0.6 |  0.7 |  0.8 |  0.9 |
> | ----------- | ---: | ---: | ---: | ---: | ---: | ---: | ---: | ---: | ---: |
> | TemPr Acc@1 |  7.4 |  9.8 | 15.4 |    - | 28.9 |    - | 37.3 |    - | 40.8 |
> | EAST Acc@1  | 20.4 | 23.3 | 25.4 | 27.0 | 28.1 | 29.0 | 29.7 | 29.9 | 29.6 |
>
> ### EK100 all noun
>
> | $\rho$      |  0.1 |  0.2 |  0.3 |  0.4 |  0.5 |  0.6 |  0.7 |  0.8 |  0.9 |
> | ----------- | ---: | ---: | ---: | ---: | ---: | ---: | ---: | ---: | ---: |
> | TemPr Acc@1 | 22.8 | 25.5 | 32.3 |    - | 43.4 |    - | 49.2 |    - | 53.5 |
> | EAST Acc@1  | 31.1 | 34.0 | 35.5 | 37.3 | 38.2 | 39.3 | 39.9 | 40.5 | 40.1 |
>
> ### EK100 all verb
>
> | $\rho$      |  0.1 |  0.2 |  0.3 |  0.4 |  0.5 |  0.6 |  0.7 |  0.8 |  0.9 |
> | ----------- | ---: | ---: | ---: | ---: | ---: | ---: | ---: | ---: | ---: |
> | TemPr Acc@1 | 21.4 | 22.5 | 34.6 |    - | 54.2 |    - | 63.8 |    - | 67.0 |
> | EAST Acc@1  | 47.2 | 52.1 | 55.0 | 56.7 | 58.4 | 59.1 | 59.5 | 59.8 | 58.4 |

---

### Official Review · Reviewer_SbPM · 2025-10-31

**Soundness:** 3
**Presentation:** 2
**Contribution:** 2
**Rating:** 6
**Confidence:** 3

**Summary:**

The paper addresses early-action prediction, where the model must recognise an action from only a partially observed video. The authors propose EAST, a training framework with three main ideas:

1.  Randomised observation-ratio sampling: During training, the method samples an observation ratio ρ ∈ {0.1,…,0.9} and splits each video into an “observed” prefix Vᵒ and an “unobserved” suffix Vᵘ. A single model is then trained to operate across all ρ, rather than training one specialised model per ρ (as prior work like TemPr typically does).

2.  Compound loss on present vs. full video: The model is trained both to classify from the partial clip (via a decoder that forecasts future features) and to classify the full clip (oracle). Optimising both prediction logits (from predicted future features) and oracle logits (from full context) is argued to make the encoder’s features discriminative even under very small observation ratios.

3. Token masking for efficiency: The authors introduce a difference-based token masking strategy that drops ~50% of spatio-temporal tokens deemed redundant across time. This reduces training memory and wall-clock time while preserving accuracy.
EAST reports strong gains on NTU60, Something-Something v2 (SSv2 and SSsub21), and UCF101, on both VideoMAE and MoViNet backbones. The paper also includes ablations: (i) with/without the decoder, (ii) with/without each loss term, (iii) different masking ratios, and (iv) per-ρ specialised models vs. one unified model. The authors claim that EAST achieves new state-of-the-art results while requiring fewer GPUs and no per-ρ retraining.

**Strengths:**

Overall, even if the ideas are incremental individually, the combination is interesting and could serve as a practical baseline for early-action prediction. Below are the main strengths of this paper.

1. Practical relevance and framing. Early action prediction is important for any system that must react before an action is complete (e.g., driving, human-robot interaction). The paper motivates this scenario well.

2. Single model across all observation ratios. Prior work often trains a different model for each observation ratio ρ. This is expensive and awkward for deployment. EAST’s randomised sampling of ρ at training time directly addresses that, and Figure 2 suggests that such a unified model generalises well across all ρ, whereas per-ρ specialists do not transfer. This is an important engineering insight for the field.

3. Compound loss improves robustness at low observation ratios. The combination of predicting from partially observed clips and supervising with an “oracle” view of the full clip encourages discriminative features even at 10%–20% observation. The ablations (Table 7) confirm that using both loss terms (L_pred and L_oracle) improves accuracy over using either alone.

4. Training-time efficiency via token masking. The masking procedure drops ~50% of “low-difference” tubelets and roughly halves peak memory usage, while maintaining accuracy (Table 6). This is a concrete, useful contribution, and the authors do report memory/TFLOP numbers and epoch times.

5. Strong experimental results. EAST achieves new state-of-the-art top-1 accuracy across a range of observation ratios on SSv2, SSsub21, NTU60, and UCF101, using both VideoMAE and MoViNet backbones. Comparisons include modern baselines like TemPr (CVPR 2023), ERA (ECCV 2022), and RACK (TIP 2023).

6. Reproducibility and ablations. The paper discloses hyperparameters, training schedule, backbone initialisations (VideoMAE pretraining, MoViNet), and even wall-clock timings. The ablation test: decoder types (direct vs autoregressive), loss design, shared vs separate classifier, masking ratios, etc.

**Weaknesses:**

1. Clarity and notation issues in the method section: the construction of Vᵒ (observed) and Vᵘ (unobserved) around a sampled ratio ρ is central, but the notation is underspecified. The text states: “The sampled clip V = Vᵒ ∥ Vᵘ consists of 2T evenly spaced frames centered at ρ·T_d,” implying |Vᵒ| = |Vᵘ| = T. However, T is not defined before use, and “centered” conflicts slightly with “concatenation of Vᵒ and Vᵘ with no gap.” This makes it unnecessarily hard to reproduce the exact temporal sampling logic.
The encoder paragraph (“Tokenizer T splits the input clip frames into N_t … tubelets of size d×p×p=2×16×16 … The transformer encoder V extracts features…”) mixes general notation and hard-coded hyperparameters in a colloquial way. Key variables (T, p, d, N_t) are introduced abruptly, and not all are defined rigorously. For a core technical section of an ICLR paper, this could be tighter.

2. Limited theoretical insight: The paper argues that combining (i) a loss on predicted future features from partial clips (L_pred) and (ii) a loss on oracle full-clip features (L_oracle) makes the encoder discriminative under all observation ratios. While plausible, this is only described qualitatively; no analysis (e.g., feature visualisations, calibration curves, entropy at low ρ) is provided.

3. Missing qualitative results: The paper does not include any qualitative timeline-style figure showing how the predicted class probabilities evolve as more of the video is seen, nor any visual comparison to ground truth. This is fairly standard in early
action anticipation literature to illustrate how “early” the model can correctly commit to an action. Adding even one such figure would greatly improve the interpretability and trustworthiness of the qualitative claims.

4. Comparisons to the most recent literature: The comparisons include TemPr (CVPR 2023), ERA (ECCV 2022), and RACK (TIP 2023), which are strong baselines. However, there are also recent approaches that model early action recognition via prototypical representations or transformer decoders and report top-1 accuracy vs. observation ratio on SSv2 / SSsub21 / EPIC-Kitchens in late 2023 (e.g., “Early Action Recognition with Action Prototypes”, a.k.a. Early-ViT-style models). These methods also aim at efficient online recognition from partial clips. EAST does not cite or compare against that family, and it would strengthen the impact claim to position EAST relative to those more recent models.

5. Evaluation metrics: The paper reports only top-1 accuracy across multiple observation ratios ρ, which is indeed standard in early action prediction work such as TemPr and other pre-2024 baselines. Still, reporting additional metrics like top-5 accuracy, AUC across ρ, or calibration / time-to-correct-prediction curves would make the evaluation more comprehensive, especially since the method is motivated by safety-critical, real-time applications.

6. Slightly overstated claims: The authors state they achieve “9× less compute,” “2× faster training,” and training on “consumer-grade hardware.” These claims are mostly motivated by:

   a.  Not needing to train nine separate models (one per ρ).

   b.  ~2× memory savings from their masking scheme (Table 6). 3. Wall-clock per-epoch measurements (80s/epoch vs 173s/epoch for TemPr on the same 4× RTX A6000 setup).

While these are promising, the paper could be more precise: RTX A6000s are still high-end GPUs; and the ‘9× less compute’ claim is inferred rather than measured in total GPU-hours. This is more of a tone issue than a scientific flaw, but worth noting.

**Questions:**

1. Clarify the temporal sampling: Please explicitly define T, T_d, Vᵒ, Vᵘ, and the operator ∥. When you say “The sampled clip V = Vᵒ ∥ Vᵘ consists of 2T evenly spaced frames centered at ρ·T_d,” does that mean |Vᵒ| = |Vᵘ| = T, with Vᵒ covering frames strictly before ρ·T_d and Vᵘ strictly after ρ·T_d, and no overlap? Or does “centered” mean you actually take a symmetric window around ρ·T_d and then label the pre-cut portion “observed” and the post-cut portion “future”? A precise definition is important for reproducibility.


2. Generalisation across ρ: You state that “training at fixed observation ratios produces models that are suboptimal at other values of ρ.” You partially support this claim with Figure 2 (specialist models vs EAST). Could you quantify how much performance drops when a ρ-specialist is evaluated out-of-distribution (e.g., a model trained at ρ=0.3 evaluated at ρ=0.7)? Are all these models trained with exactly the same backbone, number of epochs, and data seen per step, so that the comparison is fair?

3. Loss design and theory: The compound loss combines L_pred (classification on forecasted features from partial observation) and L_oracle (classification on oracle full-clip features). Can you provide intuition for why this helps so much at low observation ratios (ρ=0.1, 0.2)? For example, is the model learning to align early-frame features with full-clip features, or is it essentially acting like a teacher-student relationship inside the same network? Did you try scheduling L_oracle to zero out later in training (to avoid overfitting to full context)?

4. Token masking: The masking method selects tubelets with a large temporal difference based on an L1 distance across time. Can you include a short figure or pseudocode showing this ranking process? The current text around Eqs. (5–7) is quite dense and abruptly introduces p, d, and N_t. At inference time, do you also apply masking? If not, is there any distribution shift between masked training and unmasked inference? Does masking ever remove fast motion in the observed portion, hurting the ability to predict very early?

**Details Of Ethics Concerns:**

The paper uses standard, publicly available video datasets (SSv2, NTU, UCF101). There is no sensitive data collection or obvious misuse scenarios presented beyond typical action recognition—anticipation use cases.

---

> ### Author Response · Authors · 2025-11-23
> **Response to Reviewer SbPM**
>
> We thank the reviewer for thoroughly discussing our work. We are glad that the reviewer recognized that the proposed components in EAST work well together, and that our claims are supported by experiments.
>
> Here, we address the weaknesses singled out by the reviewer:
>
> >W1. Clarity and notation issues in the method section.
>
>    We agree to revise the method section that renders the proposed sampling strategy and will do so in the following days. To clarify, we explicitly ensure that all frames in $\mathbf{V}^o$ occur before $\rho \cdot T_d$, and that the first frame in $\mathbf{V}^u$ is the frame following the last frame in $\mathbf{V}^o$. There is no overlap between the frames in $\mathbf{V}^o$ and $\mathbf{V}^u$. In order to promote clarity, our complementing training implementation includes the sampling strategy.
>
> >W2. Limited theoretical insight.
>
> We agree that the claim can be substantiated, and we will correct it. Regarding the suggested investigation, we appreciate its thoroughness. However, we would like to respectfully note that our ablation studies (Table 7 in the main paper) already validate the contribution of each loss component to the target metric. To further address this concern, we have conducted additional experiments measuring prediction entropy, which we include in Appendix A.9. However, we don't have a high expectation of the value of proceeding the investigation. We would appreciate your thoughts on the importance of this investigation.
>
> >W3. Missing qualitative results
>
> Thank you for pointing this out. We agree that qualitative visualisation will improve the quality of the paper. We have now added the corresponding examples to the appendix, in Figure 4.
>
> >W4. Comparisons to the most recent literature
>
> To the best of our knowledge, we included all recent published work on early action prediction. Furthermore, we were aware of the Early-ViT pre-print on arXiv, but omitted their results due to missing source code. We compared against Early-ViT in the tables in the revised manuscript.
>
> >W5. Evaluation metrics
>
> We thank the reviewer for pointing out that safety-critical and time-critical applications should include metrics other than Top-1 accuracy. We re-evaluated our checkpoints to compute Top-5 accuracy as well as area under accuracy-ρ curve.
> The results for the Something-Something v2 can be seen in the table below, and will be added in revised paper in the following days.
>
> | $\rho$ |  0.1 |  0.2 |  0.3 |  0.4 |  0.5 |  0.6 |  0.7 |  0.8 |  0.9 | AUC |
> | ------ | ---: | ---: | ---: | ---: | ---: | ---: | ---: | ---: | ---: | --- |
> | Acc@1  | 25.6 | 30.1 | 34.5 | 41.6 | 49.0 | 55.2 | 59.4 | 63.0 | 64.0 | 0.4 |
> | Acc@5  | 54.3 | 59.6 | 65.0 | 72.4 | 78.9 | 83.4 | 86.0 | 87.8 | 87.9 | 0.7 |
> >W6. Slightly overstated claims
>
> We will adjust the tone regarding memory and compute savings. We perform measurements on RTX A6000s (48G vRAM). However, since EAST uses 19G vRAM per-GPU, it is possible to train on SSv2 using a single AMD RX 7900 XTX or NVIDIA RTX 3090Ti within two GPU days.
>
>
> We also prepared answers to the reviewer's questions:
>
> >Q1. Clarify the temporal sampling
>
> Please see our response to Weakness 1.
>
> >Q2. Generalisation across ρ
>
> We refer the reviewer to Table 17 in the supplement. The table contains numeric accuracy measurements visualized in Figure 2. Moreover, all the experiments use a common hyperparameter set.
>
> >Q3. Loss design and theory
>
> Our hypothesis is that $\mathcal{L}_{\mathrm{oracle}}$ serves as regularization. Under small observation ratios, there is a risk of overfitting to uninformative features within the observed frames.
>
> Backpropagating through $\mathcal{L}_{\mathrm{oracle}}$ includes a signal about the entire sequence, which benefits both the encoder and the decoder (as suggested by Table 7). The commonly used L2 loss serves a similar purpose, however it is near zero after a few training iterations and therefore does not lead to better generalization.
>
> >Q4 Token masking
>
> We will revise the text regarding the token masking algorithm, whereas the complementing source code includes the token masking implementation. Token masking is present in both training and inference. We did not measure a significant accuracy difference during unmasked inference. Here we experimented with different attention softmax temperature to account for a distribution shift, but with little change in accuracy.

---

> > ### Comment · Reviewer_SbPM · 2025-11-25
> > **Answer to Authors/rebuttal**
> >
> > Thanks for your response, for updating the tables and thanks for uploading your scripts.

---

### Official Review · Reviewer_oxAh · 2025-11-05

**Soundness:** 3
**Presentation:** 3
**Contribution:** 3
**Rating:** 6
**Confidence:** 3

**Summary:**

The paper introduces EAST (Early Action prediction Sampling strategy with Token masking), an encoder-decoder method for early action prediction. To improve classification performance, the encoder is trained considering both the previous and all the frames of the current action and employing a shared classifier. Moreover, training is performed by considering different observation ratios via sampling. This helps in having a single model able to handle various observation ratios. Finally, a token masking strategy is employed that leads to almost identical performance while only using 50% of the tokens, significantly improving the training efficiency.

**Strengths:**

- The proposed method is interesting and well-justified. The joint training of the encoder on both previous and future frames is intuitive and appears to contribute positively to the performance.
- The proposed training strategy, which considers random sampling of action observation ratio, makes the method robust against different observation ratios at test time. By using a single model for different observation ratios, the equivalent training time is significantly reduced.
- The proposed token masking strategy further improves training efficiency by using up to 50% fewer tokens without any significant performance degradation.
- Multiple ablations are considered, including the choice of the decoder, the use of shared vs separate classifiers, and various combinations of the loss functions.

**Weaknesses:**

- Although the introduction highlights the importance of inference time efficiency in various applications, only the training time and efficiency are discussed in the text. It is crucial to provide more details regarding the inference time of the proposed method.
- Token masking is frequently considered in similar applications. Hence, although it most certainly helps in reducing the training efficiency, it cannot be considered a main contribution introduced in this work.
- Related to both previous comments, masking at inference time is not considered in this work. It would be interesting to explore also this direction, which could contribute to the reduction of the processing time also during inference.

**Questions:**

- What is the inference time of the proposed method?
- Have inference time masking strategies been considered?
- Why only sample at fixed increments of 0.1 and not uniformly in (0,1)?

---

> ### Author Response · Authors · 2025-11-23
> **Response to Reviewer oxAh**
>
> We greatly appreciate the comments regarding the proposed methodology and the experimental setup. We comment on the weaknesses as follows:
>
> >Details regarding the inference time
>
> Thank you for pointing this out. We acknowledge that we did not discuss inference time in the manuscript and we will include it in the revised version. Since EAST uses token masking in both training and inference, the execution time on a single RTX A6000 is 12 milliseconds, while e.g. TemPr takes 78.5ms.
>
> >Token masking
>
> We believe this is a novelty, since we demonstrate that token masking is indeed useful in early action prediction as well. This work is the first to enable memory efficient training using ViT encoders in early action prediction.
>
> >Masking at inference time
>
> We will make the manuscript clear in this regard: EAST uses token masking in both training and inference.
>
> Next, we answer the questions:
> - The inference time is around 12 milliseconds per input video clip.
>
> - EAST uses difference masking $\mathcal{M}_k^{\text{d}}$ in both training and inference. This is different when compared to VideoMAE, since their supervised fine-tuning does not use any masking. The original submission does not clearly explain that. We will revise the manuscript accordingly.
>
> - We did not consider sampling $\rho$ uniformly when preparing the original submission. This is a good suggestion. We performed this experiment and present the results here. Note that there are no significant differences in the results since the originally proposed sampling covers most sampling combinations. We present the results on SSv2 in the table below.
>
> | $\rho$                      |  0.1 |  0.2 |  0.3 |  0.4 |  0.5 |  0.6 |  0.7 |  0.8 |  0.9 |
> | --------------------------- | ---: | ---: | ---: | ---: | ---: | ---: | ---: | ---: | ---: |
> | $\\{0.1, \ldots, 0.9\\}$ Acc@1 | 25.6 | 30.1 | 34.5 | 41.6 | 49.0 | 55.2 | 59.4 | 63.0 | 64.0 |
> | $\mathcal{U}(0, 1)$ Acc@1   | 25.5 | 30.0 | 34.3 | 41.3 | 48.6 | 55.2 | 59.7 | 62.9 | 63.7 |

---

### Author Response · Authors · 2025-11-23
**Comment to the reviewers and ACs**

**Implementation.**
We highlight the implementation details the reviewers were most interested in:
- Token masking implementation is available in `engine_for_finetuning.py`, lines 61-105.
- Frame sampling strategy is available in `ssv2.py`, lines 272-281. The resulting `all_index_` holds the observed frame indices in the first half. For reference, the code in lines 260-269 holds the original sampling used by VideoMAE in action recognition.

**Source code**. A link to an anonymous source code repository is available [here](https://anonymous.4open.science/r/EAST-5776).
Please note that we keep the comment visible to reviewers and ACs only.

---

### Author Response · Authors · 2025-11-23
**Comment to the Reviewers**

We are thankful that the reviewers took the time to thoroughly consider our manuscript!
We are glad that all the reviewers have highlighted the same strengths of our work.
The reviews are indeed helpful in making our submission more meaningful and understandable.
We conducted additional experiments to address the reviewers' questions.
Due to limited computational resources and finite time to address the reviews, some experiments are still in progress.

We hope that our discussion continues after the initial responses.
In the upcoming days, our goal is to submit a revised manuscript that follows the reviewer suggestions.

---

### Author Response · Authors · 2025-12-03
**Message to new AC**

We sincerely thank all the reviewers for their time, thoughtful insights and constructive feedback. Their comments significantly strengthened our work and encouraged us to include additional experiments. We hope this revision provides additional clarity. This comment summarizes the reviews, enumerates the updates to the original manuscript, and points out erroneous claims from a review.

Although we are unable to further participate in valuable discussions, we remain appreciative for the expertise and new perspective the reviewers brought to our attention.

**Key Reviewer Concerns Addressed**

A key question raised by reviewers oxAh and 9qPd was the inference time masking. The revised manuscript now clearly states that EAST uses masking in both training and inference. Also, the paper now includes inference time measurements that are compared with the current state-of-the-art.

We conducted all additional experiments suggested by the reviewers, including:
- sampling $\rho$ uniformly in (0,1), showing similar performance to our proposed sampling strategy (oxAh)
- experiment with MAR masking, showing that our masking strategy remains superior (9qPd)
- evaluation on additional dataset (EPIC-KITCHENS-100) (9qPd)
- additional comparison between VideoMAE and $\text{EAST}_{\varepsilon}$ on UCF101, confirming that VideoMAE only performs competitively around observation ratio it was trained for (mSTD)

We also added qualitative examples in the revised paper, as requested by reviewer SbPM.
All changes in the revised manuscript are in blue.

**Reviewer Validation**

We appreciate that reviewers find our method is:
- "interesting and well-justified" and "robust against different observation ratios at test time" (oxAh),
- that "training time is significantly reduced" (oxAh)
- "Practical relevance and framing" about early action prediction (SbPM)
- "Strong experimental results", "Reproducibility and ablations"  (SbPM)
-  "In-depth analysis" and " Comprehensive evaluation" (9qPd)
- "Very impressive evaluation results" (mSTD)
- that overall approach bears a good balance of simplicity, optimization strategy and ratio-agnostic design (mSTD)

**Motivating discussions**

We were especially interested to hear more from Reviewer mSTD, since the reviewer agreed it was interesting that action prediction models fail on low observation ratios. We have confirmed the trend on another dataset (UCF-101) after the discussion was closed. The newer insights also address the questions in weakness D posted by Reviewer mSTD.

**Reviewer 9qPd**

We would also like to kindly point out the original review by Reviewer 9qPd.
Our original response was written to open discussion with the reviewer, but the reviewer did not reply before the OpenReview incident.
The review contains the following errors:
- W2. "Outdated baselines:", "analysis on temporally challenging datasets (e.g., Something-Something)": We believe a misunderstanding occurred, as the original version of our paper already contains the analyses on the Something-Something dataset.
- W4. "Limited masking comparison": The reviewer stated that we omit VideoMAE masking, which is incorrect as VideoMAE uses random masking that we tested in our original manuscript. We have also included MAR masking experiments, as suggested by the reviewer.

We also find that "W1. Weak motivation & significance" is in contrast with the evaluations of all other reviewers that said the method is "well-justified" (oxAh), practically relevant and "important" (SbPM) and "straightforward and easy to implement" (mSTD, confidence 4).

---

### Meta-Review · Area_Chair_EMFW · 2026-01-07

**Summary:**

Reviewer oxAh (rating: 6) pointed out that no inference-time analysis was provided, and token masking is not novel.

Reviewer SbPM (rating: 6) was concerned about the limited theoretical and qualitative analysis, and the missing comparisons to recent early action models.

Reviewer 9qPd (rating: 2) found that the paper is poorly motivated, the comparison is out-of-date, and the evaluation is insufficient.

Reviewer mSTD (rating: 6) has a strong concern about the usage of the token masking module and requested clarification about the protocol difference against Tempr.

**Reviewer Concerns:**

The AC has checked the review and the response, and concluded that the authors have provided an adequate rebuttal to address all the comments one-by-one. In particular, the concerns from Reviewer 9qPd, who gave the only negative rating, was well addressed by incoparating more experimental results.

**Reviewer Scores:**

Reviewer 9qPd increased the rating to 6, and the other reviewers keep the positive rating.

---

### Decision · Program_Chairs · 2026-01-26

Accept (Poster)